https://doi.org/10.1038/s41467-020-19085-1　　**OPEN**

# Non-dispersive infrared multi-gas sensing via nanoantenna integrated narrowband detectors

Xiaochao Tan[1], Heng Zhang[1], Junyu Li[1], Haowei Wan[1], Qiushi Guo[2], Houbin Zhu[3], Huan Liu[1] & Fei Yi [1✉]

Non-dispersive infrared (NDIR) spectroscopy analyzes the concentration of target gases based on their characteristic infrared absorption. In conventional NDIR gas sensors, an infrared detector has to pair with a bandpass filter to select the target gas. However, multiplexed NDIR gas sensing requires multiple pairs of bandpass filters and detectors, which makes the sensor bulky and expensive. Here, we propose a multiplexed NDIR gas sensing platform consisting of a narrowband infrared detector array as read-out. By integrating plasmonic metamaterial absorbers with pyroelectric detectors at the pixel level, the detectors exhibit spectrally tunable and narrowband photoresponses, circumventing the need for separate bandpass filter arrays. We demonstrate the sensing of $H_2S$, $CH_4$, $CO_2$, $CO$, $NO$, $CH_2O$, $NO_2$, $SO_2$. The detection limits of common gases such as $CH_4$, $CO_2$, and $CO$ are 63 ppm, 2 ppm, and 11 ppm, respectively. We also demonstrate the deduction of the concentrations of two target gases in a mixture.

[1] School of Optical and Electronic Information and Wuhan National Research Center for Optoelectronics (WNLO), Huazhong University of Science and Technology, Wuhan 430074, China. [2] Department of Electrical Engineering, Yale University, New Haven, CT 06511, USA. [3] School of Physics, Shandong University, Jinan 250100, China. ✉email: feiyi@hust.edu.cn

The mid-infrared (mid-IR) spectral range (wavelength $\lambda \sim 2$ to 20 μm) is known as the "molecular fingerprint" region, where a wide variety of gas molecules exhibit highly characteristic rotational or vibrational transition bands[1–5]. Notably, in the mid-IR region, the absorption strengths of molecular transitions are typically 10–1000 times greater than those in the visible or near-IR region[6]. As such, mid-IR spectroscopic gas sensors can be employed to uniquely identify and quantify the presence of substances with high sensitivity and selectivity. Non-dispersive infrared (NDIR) spectroscopy is one of mid-IR spectroscopic gas sensors that analyzes gases based on their characteristic absorption wavelengths in the mid-IR caused by their molecular vibrations[1,7], which can find profound applications in traced gas sensing[8,9], breadth analysis[10,11], environmental monitoring[12,13], to name a few.

In conventional NDIR gas sensor, the light source is broadband and not pre-filtered. When the light beam containing a wide range of wavelengths passes through and interacts with the sample gases in a chamber, only a portion of the optical energy is absorbed by the gases at their characteristic absorption wavelengths[14]. To analyze the concentration of a target gas, a bandpass optical filter is typically added before the detectors to eliminate all unwanted wavelengths in the light beam and only allow the characteristic absorption wavelengths of the gas to reach the detector. In other words, the spectral selectivity in conventional NDIR architecture is enabled by the added filters, rather than the detectors (see Supplementary Note 1).

In order to analyze several target gases in a mixture at the same time, one can simply implement multiple pairs of "bandpass filter + optical detector" in the NDIR gas sensor[15,16]. However, this scheme greatly increases the cost, the system complexity as well as the operating time, especially when the number of target gases is large[17,18]. Such challenge is fundamentally rooted in the lack of spectral selectivity of most commercially available mid-IR detectors. One way to avoid the need for separate optical filters is to introduce pixel-level spectral selectivity to mid-IR detectors by integrating plasmonic metamaterial absorbers (PMAs) onto the detector pixel. PMA is composed of arrays of metallic plasmonic resonators that can selectively absorb a certain spectral band of light and, therefore, can be regarded as nanoscale absorption filters[19–23].

Following this idea, we propose a new NDIR architecture in which an array of narrowband PMA integrated pyroelectric elements are used to spectrally resolve the absorption of multiple gases at the same time[24]. By tuning the geometry of metallic plasmonic resonators, the central detection wavelength of each element can be independently adjusted to match the characteristic absorption bands of different target gases. The multiplexed sensing platform can thus be used to analyze multiple target gases in a mixture with significantly reduced device footprint and operating time. Leveraging the proposed gas sensing platform, we demonstrate the sensing of 8 different gases: $H_2S$, $CH_4$, $CO_2$, $CO$, $NO$, $CH_2O$, $NO_2$, $SO_2$, with the detection limits of 489, 63, 2, 11, 17, 27, 54, 104 ppm, respectively. We also demonstrate that the concentrations of two target gases in a mixture can be deduced from the voltage responses of two narrowband detectors. Although the sensing platform in its current form is still bulkier than the commercial NDIR sensors, we believe that by reducing the thickness of the pyroelectric elements and improving the quality factors of the narrowband PMA, integrated multiplexed gas sensors with centimeter long sizes can be achieved.

## Results

**Principle of operation.** The schematic diagram of the proposed NDIR multiplexed gas sensing platform, which is composed of three parts: the broadband light source, the gas cell and the multiplexed sensor with necessary focusing optics is plotted in Fig. 1a. An example of packaged multiplexed pyroelectric sensor with different detection wavelengths for spectral sensing is shown

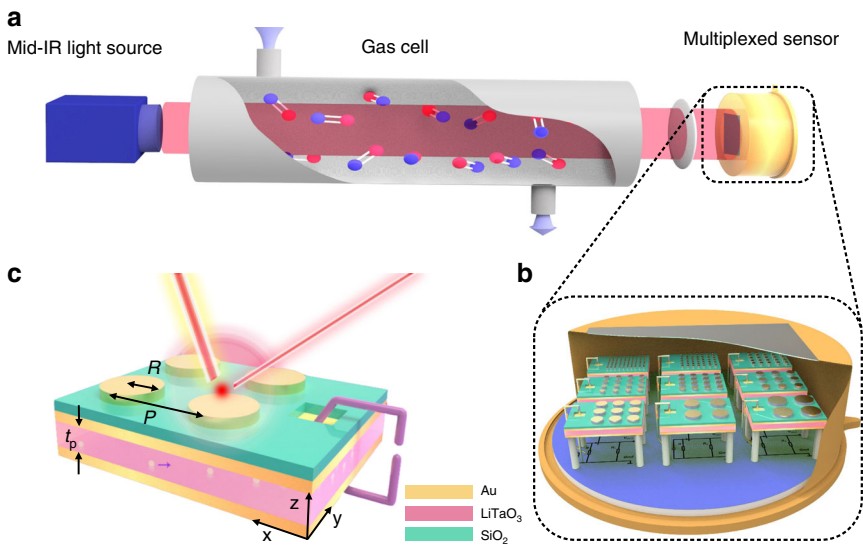

**Fig. 1 An overview of the proposed non-dispersive infrared (NDIR) architecture using metamaterial enabled narrowband pyroelectric detectors. a** A schematic diagram of the gas sensing system based on the proposed NDIR architecture with an array of narrowband PMA integrated pyroelectric elements used as the spectral sensor. The system is composed of three parts: the broadband light source, the gas cell, and the multi-element sensor together with necessary focusing optics. **b** The joint package of the multiple pyroelectric elements with different detection wavelengths. **c** The device geometry of the narrowband detection element. From the top to the bottom are: the Au nanodisk antenna, the silicon dioxide spacer, the gold backplate that is also used as the top electrode of the pyroelectric element, the lithium tantalate (LT) substrate, and the gold bottom electrode. The length of the smallest repeatable unit is the periodicity $P$ and the radius of the nanodisk is $R$. The area size of each absorber is $1 \times 1$ mm and the thickness $t_p$ of the LT substrate is 75 μm. To provide electrical access to the gold backplate buried underneath the silicon dioxide spacer, a window area beside each absorber is opened by removing the spacer.

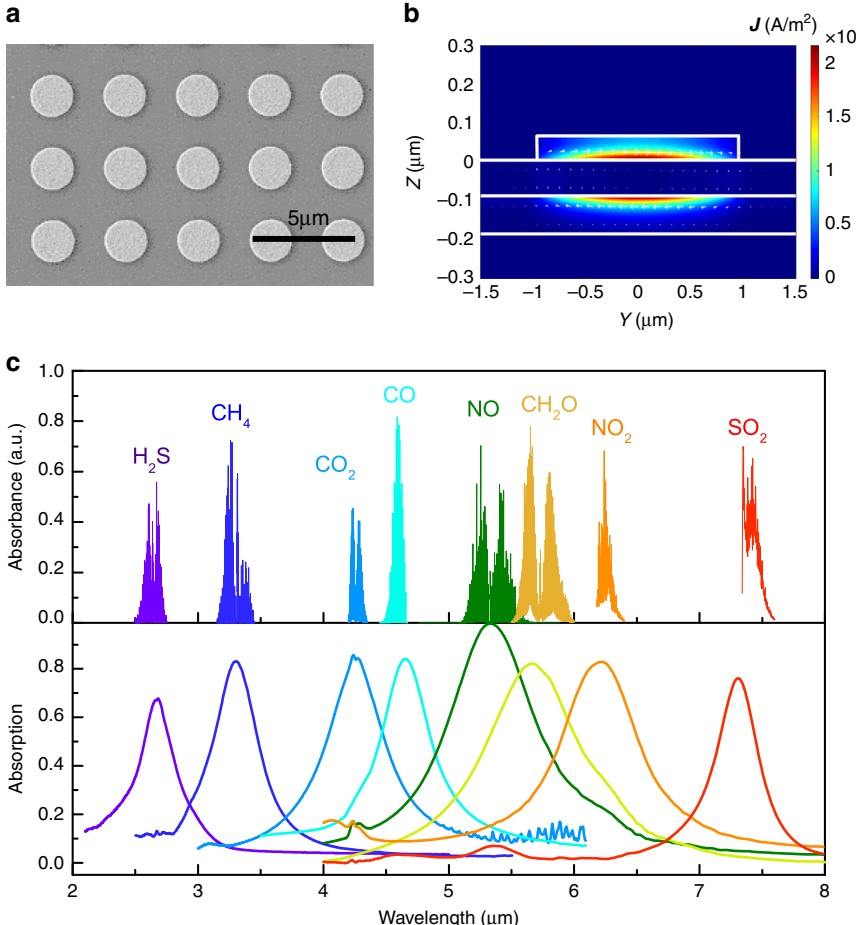

**Fig. 2 The spectral and near-field properties of the plasmonic metamaterial absorbers. a** The scanning electron microscope (SEM) image of the gold nanodisk antenna array. **b** The distribution of light-induced current density magnitude |**J**| and current density vector **J** in the YZ cut-plane of the MIM absorber at $\lambda_{peak} = 5.73\,\mu m$, obtained by COMSOL, a finite element method-based solver. **c** The measured absorption spectra of 8 fabricated MIM absorbers compared to the infrared absorption bands of eight target gases: $H_2S$, $CH_4$, $CO_2$, CO, NO, $CH_2O$, $NO_2$, $SO_2$. See Supplementary Note 7 for the absorption spectra in the full wavelength range. Source data are provided as a Source Data file.

in Fig. 1b. The absorption filter is essentially a metal-insulator-metal (MIM)-based metamaterial absorber that consists a top layer of gold nanodisk antennas, a $SiO_2$ spacer, and a gold backplate. The absorber is directly fabricated on top of a commercially available thin lithium tantalate (LT) substrate with pre-deposited gold electrodes on the top and bottom surfaces. The top gold electrode of the LT substrate is also used as the gold backplate of the MIM absorbers for simplicity. LT as the sensing material offers a very broadband infrared response, enough to cover the characteristic absorption bands of typical gases, and a high pyroelectric coefficient[25]. For simplicity, here we cut the LT substrate with integrated MIM absorbers into separate elements with different spectral responsivity. Apparently, one can increase the number of elements in the package to monitor more gases with different spectral response. Figure 1c illustrates the operating principle of one narrowband detection element. The gold nanodisk antennas serve to resonantly absorb the mid-IR radiation in a narrowband fashion and convert absorbed optical energy into heat[26], which elevates the temperature of the LT substrate. The resulting temperature increase $\Delta T$ in turn causes the LT layer to generate the pyroelectric readout current $\Delta I_{out}$[27,28], which is then converted to readout voltage $\Delta V_{out}$ by the readout electronics.

**Design of plasmonic metamaterial absorbers.** To sense multiple target gases, the spectral absorption of the narrowband detectors

should be designed to match the characteristic absorption bands of different target gases in the mid-IR. Figure 2a shows the scanning electron microscope (SEM) image of the fabricated gold nanodisk antenna array (see Supplementary Note 2 for the details about the design of the MIM absorbers for 8 target gases, and Supplementary Note 3 about the fabrication and package of the narrowband detectors). To reveal the microscopic picture of resonant light trapping and dissipation of the optical energy, we need to look at the distribution of the optically induced currents in the absorber. Figure 2b plots the local distribution of the light-induced current density magnitude |**J**| and current density vector **J** in an MIM absorber at its resonant wavelength $\lambda_{peak} = 5.73\,\mu m$. Under the excitation of the y-polarized (Cartesian coordinate system in Fig. 1c) plane wave, the induced local currents oscillating along the y-direction are maximized in the center region of the nanodisk antenna and decrease in magnitude towards the edges of the antenna[29], resulting in net electric charges and enhanced local electric fields near the antenna edges. The ohmic loss caused by the oscillating currents is mainly distributed at the lower surface of the antenna and the upper surface of the gold backplate, which serves to heat up the LT underneath. Figure 2c shows the measured absorption spectra of the 8 fabricated MIM absorbers and the characteristic IR absorption spectra of their target gases, whose absorption bands are far away from each other. It ought to be noted that by optimizing the antenna

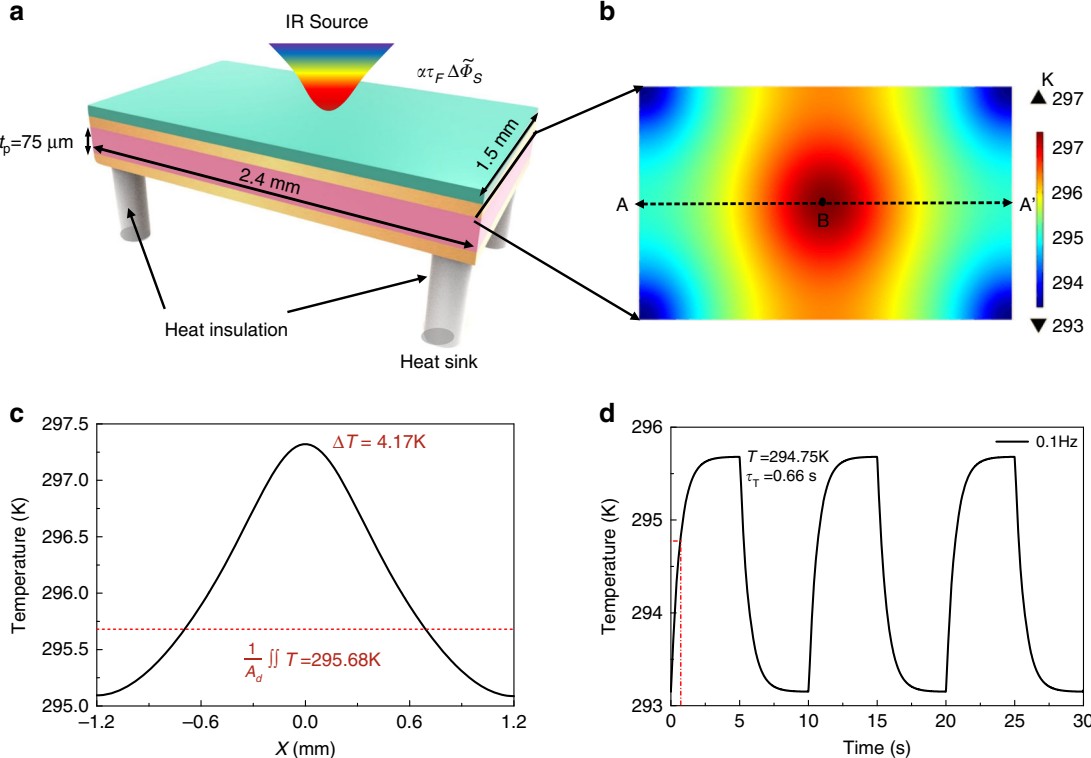

**Fig. 3 The thermal analysis of the pyroelectric detector. a** The schematic diagram of the suspended LT substrate used for calculating the steady-state temperature distribution and the dynamic temperature change in the time-domain. **b** The steady-state temperature distribution at the surface of the LT layer. **c** The steady-state temperature distribution along the cut line A-A'. The red dotted line indicates that the average temperature is 295.68 K. **d** The time-domain average temperature at the upper surface of the LT substrate assuming that the power is modulated by a square wave with the frequency of 0.1 Hz. Source data are provided as a Source Data file.

structure and the pattern of the array[30], or replacing the metallic antennas with dielectric antennas[31], the line-widths of the narrowband detectors can be further reduced, which in turn, improves the sensor selectivity.

**Heat generation in the detector.** Since the sensitivity of the NDIR sensor to the gases is strongly correlated to the mid-IR responsivity of each narrowband detector element, it is necessary to accurately model the photothermal and the temporal response of the detector element. We used the heat transfer module in COMSOL Multiphysics, a finite element method based solver, to calculate the steady-state distribution of the temperature increase $\Delta T$ in the PMA integrated LT detector caused by the dissipated electromagnetic energy. Figure 3a shows the heat transfer model of the suspended LT substrate. The four corners of the LT substrate are suspended by silicon posts. The simulated steady-state distribution of the temperature across the upper surface of the LT layer is shown in Fig. 3b. We also plot the steady-state temperature distribution along the cut line A-A' in Fig. 3c. The maximum steady-state temperature at the center point B is found to be $T_{center} = 297.32$ K. The average value of the steady-state temperature $T_{average}$, defined as the steady-state temperature distribution integrated over the upper surface of the LT substrate and divided by the area size of the upper surface of the LT substrate, is then calculated be 295.68 K, as shown by the red dotted line in Fig. 3c. We also plot the increase of the average temperature $T_{average}(t)$ in the time-domain in Fig. 3d. The thermal time constant $\tau_T$ is found to be 0.66 s. (The temperature change as a function of the modulation frequency of the optical chopper is shown in Supplementary Note 4).

**Photoresponse of the narrowband detectors.** In order to assess the spectral response of the narrowband detectors, we measured the IR absorption spectra (Fig. 4a black curve) and the wavelength-dependent voltage responses (Fig. 4a red curve) by using a frequency-tunable quantum cascade laser (QCL). When the radius of Au nanodisk is 0.94 μm and the periodicity is 3 μm, the detector has a narrowband absorption spectrum peaked at 5.52 μm with a full width at half maximum (FWHM) of 670 nm. Importantly, the wavelength-dependent voltage responses of the detectors reproduce the IR absorption spectra of the PMAs very well. Figure 4b shows the voltage response of the detector as a function of the modulation frequency. It is found that when the modulation frequency is 7 Hz, the output voltage of the detector drops to 70.7% (3 dB) relative to the output voltage at the frequency of 4 Hz. Therefore, the modulation frequency should be set below 7 Hz. Moreover, the inset is the dynamic response of the narrowband detector (5.52 μm) in the time-domain at the modulation frequency of 5 Hz recorded by an oscilloscope (Tektronix, DPO2024B). The average measured voltage response at the modulation frequency of 5 Hz is 90 V W$^{-1}$. Factors that cause the measured responsivity to be lower than the calculated responsivity include: (1) The heat conduction between the LT element and the printed circuit board containing the impedance matching circuit is more significant than the calculated case. (2) The actual spot size of the optical beam arriving at the LT element can be smaller than the area size of the LT element (see Supplementary Note 5 for the theoretical calculation of the voltage response).

To determine the noise equivalent power (NEP), which is a direct measure of the smallest optical power that can be measured by the detector, we need to evaluate the noise in the detector, or the fluctuation in the output voltage, without any input optical

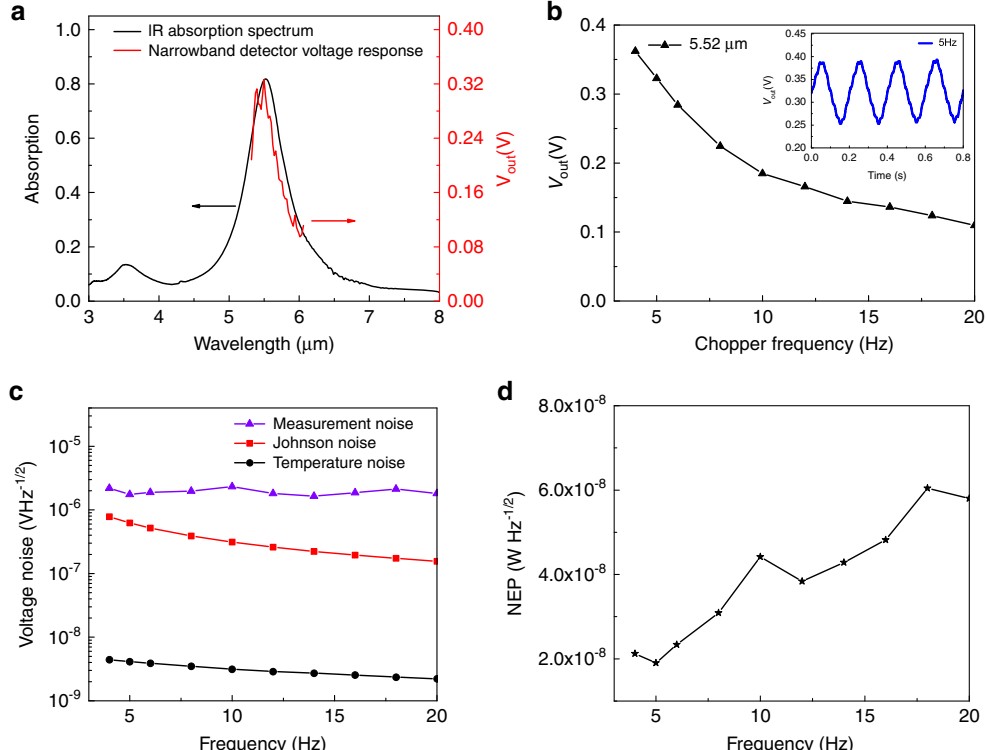

**Fig. 4 Photoresponse of detector. a** The measured voltage responses of one narrowband detector as a function of the wavelength of the input beam, compared with the absorption spectra of the integrated PMAs. The input beam is from a tunable quantum cascade laser and is modulated by an optical chopper at the frequency of 5 Hz. **b** The voltage response of the detector as a function of the chopping frequency. The inset is the time-domain voltage response of the narrowband detector with a peak wavelength of 5.52 μm. **c** The measured fluctuation (total noise) in the output voltage of the detector without input optical power, and the calculated values of two major sources of noise: temperature noise and Johnson noise as a function of the frequency. **d** The noise equivalent power calculated from the measured total noise and the responsivity of the narrowband detector (5.52 μm). Source data are provided as a Source Data file.

power. The two major sources of noise here are: (1) thermal fluctuation noise $\tilde{u}_{NT}$ accounting for the random fluctuations in temperature due to the statistical nature of the heat exchange between the suspended LT detector and the supporting pins on the circuit board; (2) Johnson noise $\tilde{u}_{NR}$ originating from the thermal agitation of the electrons inside the electrical conductors at equilibrium. The LT element is a capacitive structure with loss resistance, and its Johnson noise exhibits a frequency dependence due to the product of the loss resistance and the capacitance. Figure 4c plots the calculated thermal fluctuation noise $\tilde{u}_{NT}$ and Johnson noise $\tilde{u}_{NR}$ (see Supplementary Note 5 for the theoretical calculation of the noises). The total voltage noise of the detector at different modulation frequency is also plotted in Fig. 4c. The measured noise level agrees well with the calculated value when the modulation frequency is below 20 Hz. As presented in Fig. 4d, At the modulation frequency of 5 Hz, the NEP is found to be $1.90 \times 10^{-8}$ W Hz$^{-1/2}$. From the comparison of the narrowband detectors with commercial LT detectors in Supplementary Note 8, it is seen that the performance of the narrowband detectors enabled by plasmonic metamaterial absorber developed in this work is comparable to the performance of the commercial LT detectors that use metal black coating as the IR absorber.

**The NDIR experiment on single-target gas**. We then constructed an NDIR system to examine the performances of the fabricated narrowband detectors in gas sensing. As shown in Fig. 5a, the target gas is mixed with pure nitrogen gas and injected into the gas cell. The gas cell used in this work is a White type multipass cell with an effective optical length of 5 m. See

Supplementary Note 9 for the arrangement of the components in the NDIR system. A collimated light beam from a SiC broadband infrared source is modulated by an optical chopper before it entered the gas cell. The light beam passing through the gas cell is then focused onto the sensing area of the packaged pyroelectric detector. The generated pyroelectric current is converted into the output voltage by the integrated circuit (IC) and the output voltage is then measured using the lock-in amplifier. Figure 5b presents the results of the single-target gas sensing experiment for eight gases: $H_2S$, $CH_4$, $CO_2$, $CO$, $NO$, $CH_2O$, $NO_2$, and $SO_2$. Since the absolute value of the detector output voltage may vary among each experiment, we use the relative change in the output voltage $\Delta V/V_0$ to represent the voltage response of the detector. $V_0$ is the initial voltage output of the detector when the chamber is filled with pure nitrogen gas. $V$ is the voltage output of the detector when the target gas is mixed into the chamber and $\Delta V \equiv V - V_0$ is the change in the output voltage caused by the target gas. To fit the measured detector response $\Delta V/V_0$ as a function of the target gas concentration, we use a modified Beer–Lambert equation[32–36]:

$$\Delta v/v_0 = \text{span} * \left( e^{-\kappa l x^c} - 1 \right). \tag{1}$$

The modified version of the Beer–Lambert Law is required by the practical considerations in the NDIR implementation. The coefficient span accounts for the fact that not all the IR radiation that impinges upon the detector is absorbed by the gas, even at high concentrations. The value of span ranges from 0 to 1 because of the optical filter bandwidth and the fine structure of the absorption spectra. The coefficient $\kappa$ represents the effective absorption coefficient of the gas. See Supplementary Note 10 for

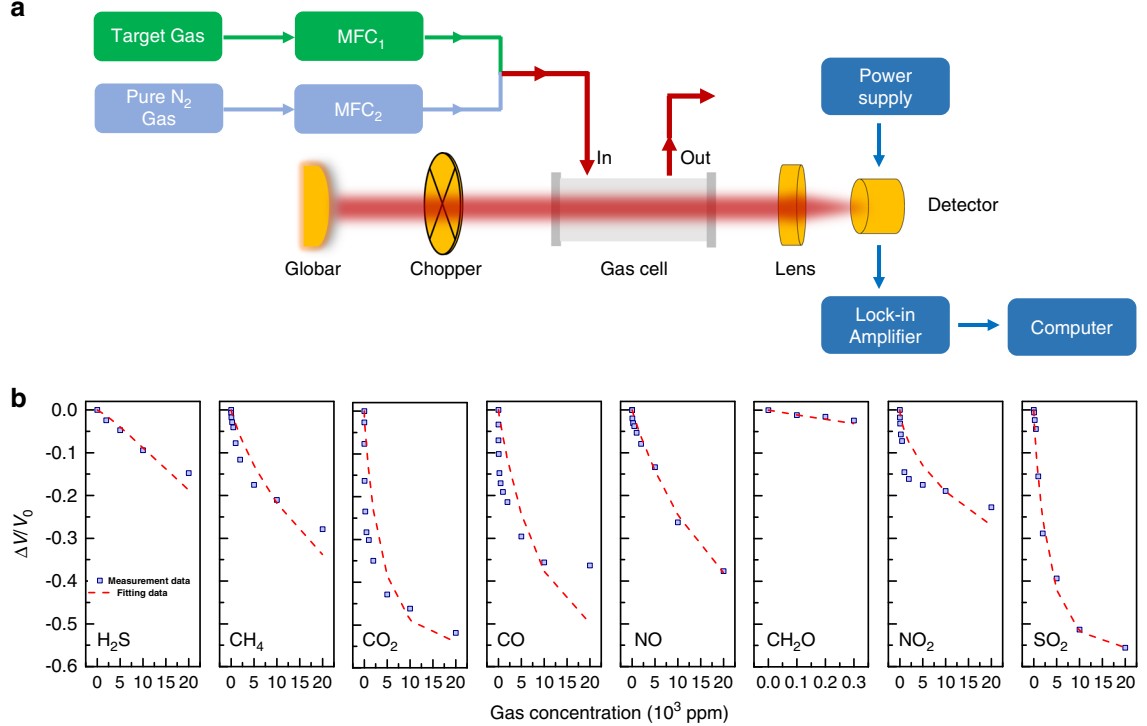

**Fig. 5 The NDIR experiment on single-target gas using the narrowband pyroelectric detectors. a** The gas sensing system. **b** The voltage responses of the narrowband detectors as a function of the concentrations of the target gases. The purple squares stand for the measured data while the red dashed lines are the fitting curves based on the Beer–Lambert law. Source data are provided as a Source Data file.

the details about the calculation of $k$ from the gas absorption lines obtained from the HITRAN database. $l = 5$ m represents the optical path length of the White type multipass cell; $x$ is the gas concentration. The parameter $c$ is added into the power term as a linearization coefficient to account for the variations in the optical path length and light scattering for accurately fitting the equation to the actual absorption data. In practice, the parameters span and $c$ are fitting parameters that are adjusted to match the fitting curves to the measured data as close as possible. See Supplementary Table 7 in Supplementary Note 12 for the detection limit of eight target gases in single-target gas measurement.

**The NDIR experiment on mixed target gases.** When there are multiple target gases in the gas cell, one can use multiple narrowband detectors to measure the gas mixture in the cell and back-calculate the concentration of each target gas based on the measured responses of the detectors. For example, assuming there are $M$ target gases in the gas cell, and $N$ narrowband detectors are used for measurement. The voltage response of the $i$th detector $D_i \equiv \Delta V_i / V_{0i}$ is related to the concentration of each target gas by:

$$D_i = \sum_{j=1}^{M} \text{span}_{ij} * \left(e^{-k_{ij}lx_j^{c_{ij}}} - 1\right). \quad (2)$$

Here, $i$ is the number of the detector and $j$ is the number of the target gas. The parameter $k_{ij}$ are calculated using Supplementary Equation (8) in Supplementary Note 10. The parameters $\text{span}_{ij}$ and $c_{ij}$ are fitting parameters that account for the contribution of the $j$th target gas to the response $D_i$ of the $i$th detector.

Taking a simple case where two target gases are measured by two narrowband detectors ("two-gas-two-detector") as an example. The voltage responses of the two detectors are related to the concentrations of the two target gases by:

$$D_1 = \text{span}_{11} * \left(e^{-k_{11}lx_1^{c_{11}}} - 1\right) + \text{span}_{12} * \left(e^{-k_{12}lx_2^{c_{12}}} - 1\right), \quad (3)$$

$$D_2 = \text{span}_{21} * \left(e^{-k_{21}lx_1^{c_{21}}} - 1\right) + \text{span}_{22} * \left(e^{-k_{22}lx_2^{c_{22}}} - 1\right). \quad (4)$$

We chose CO (gas 1) and $SO_2$ (gas 2) as the two target gases for the mixed-gas experiment. The characteristic absorption wavelengths of the two gases are: $\lambda_1 = 4.67$ μm and $\lambda_2 = 7.35$ μm, respectively. Correspondingly, the detection wavelengths of the two detectors are tuned to be 4.67 μm (detector I) and 7.35 μm (detector II), respectively. To determine the values of $\text{span}_{ij}$, and $c_{ij}$, we first performed four single-target gas measurements: (1) CO measured by detector I; (2) $SO_2$ measured by detector I; (3) CO measured by detector II; (4) $SO_2$ measured by detector II, as shown in Fig. 6a. By fitting the measured detector responses, we can obtain the values of $\text{span}_{ij}$, and $c_{ij}$. When the values of $\text{span}_{ij}$, $c_{ij}$ and $k_{ij}$ are determined, the mathematical model of the "two-gas-two-detector" problem is established. We then performed four mixed-gas experiments to verify the mathematical model: (1) CO with varying concentration and $SO_2$ with fixed concentration measured by detector I; (2) $SO_2$ with varying concentration and CO with fixed concentration measured by detector I; (3) CO with varying concentration and $SO_2$ with fixed concentration measured by detector II; (4) $SO_2$ with varying concentration and CO with fixed concentration measured by detector II, respectively. In each mixed-gas experiment, the fixed concentration is chosen to be 7500 ppm. The measured responses of detector I and detector II are plotted in Fig. 6b using purple squares. We also plot Fig. 6b the calculated detector responses based on Eq. (3) and Eq. (4) using red dashed lines (see Supplementary Note 11 for the model parameters of the "two-gas-two-detector" problem). It is seen that the detector responses predicted by Eq. (3) and Eq. (4) agree well with the measurements, which confirms the effectiveness of the mathematical model. See Supplementary Note 11 about using mathematical model to work out the gas concentrations $x_1$ and $x_2$ from the detector responses $D_1$ and $D_2$.

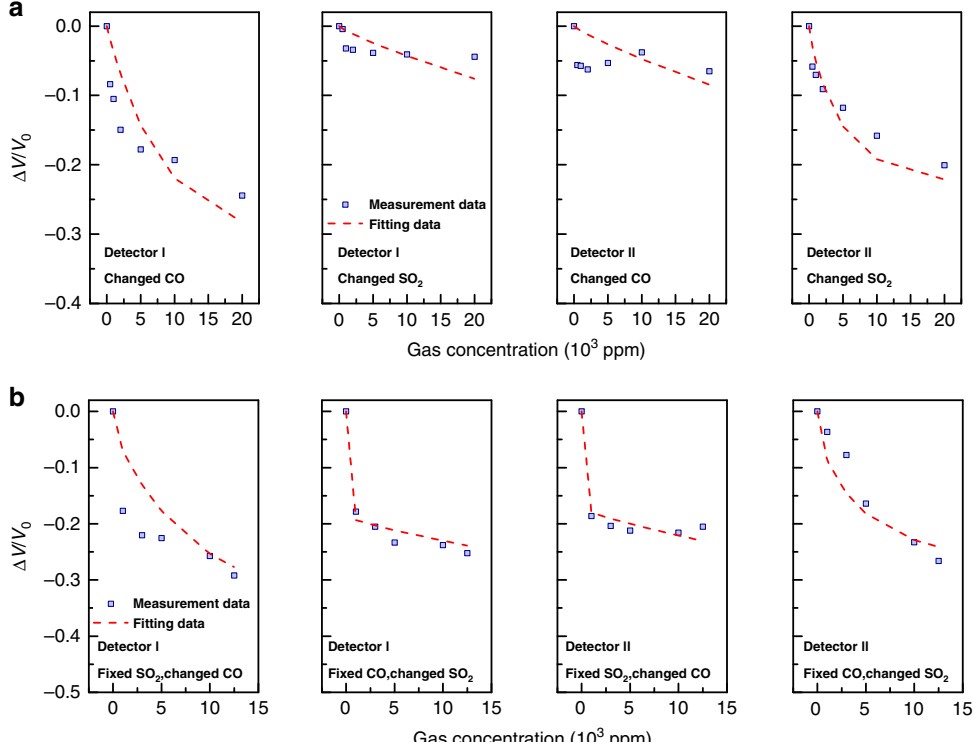

**Fig. 6 The NDIR experiment on two target gases measured by two narrowband detectors. a** The detector responses of four single-target gas measurements: CO measured by detector I; $SO_2$ measured by detector I; CO measured by detector II; $SO_2$ measured by detector II, respectively. **b** The detector responses of four mixed-gas experiments: fixed $SO_2$ concentration and varying CO concentration measured by detector I; fixed CO concentration and varying $SO_2$ concentration measured by detector I; fixed $SO_2$ concentration and varying CO concentration measured by detector II; fixed CO concentration and varying $SO_2$ concentration measured by detector II, respectively. In each mixed-gas experiment, the fixed concentration is chosen to be 7500 ppm. The purple squares stand for the measured detector responses while the red dashed lines are the calculated detector responses based on the mathematical models provide by Eq. (3) and Eq. (4). Source data are provided as a Source Data file.

## Discussion

In summary, we have presented the design, fabrication and measurement of narrowband pyroelectric detectors that are enabled by directly integrating plasmonic metamaterial absorbers onto the sensing area of lithium tantalate elements as on-chip absorption filters. The detection wavelength, or the peak absorption wavelength of the integrated absorber can be tuned to cover the full mid-infrared spectrum by varying the design of the metamaterials. We fabricated eight narrowband detectors whose detection wavelengths are aligned with the characteristic absorption wavelength of eight target gases: $H_2S$, $CH_4$, $CO_2$, CO, NO, $CH_2O$, $NO_2$, $SO_2$, and implement the detectors in a home-built NDIR system to measure the gas concentrations. The mathematical model for a simple case in which two target gases (CO and $SO_2$) in a mixture are measured by two narrowband detectors is also provided. The model can successfully predict the measured responses of the two detectors based on the concentrations of the two target gases. We also build a computer program based on the mathematical model and demonstrate the deduction of the gas concentrations $x_1$ and $x_2$ from the detector responses $D_1$ and $D_2$. The presented work thus goes beyond the conventional NDIR gas sensor in multiplexed gas sensing by circumventing the need for multiple pairs of "bandpass filter + optical detector".

Since pyroelectric materials have long been used to construct low-cost and uncooled detectors throughout the infrared spectrum, the demonstrated narrowband detectors find immediate commercial applications such as the detection of fire flame and human bodies based on their infrared absorption features and non-contact temperature measurement. In particular, pyroelectric

detectors based on LT have been massively produced using standard tools for making IC chips[37]. The nanoantenna array was also fabricated using regular fabrication processes in IC industry such as electron beam lithography, electron beam evaporation, and metal lift-off. In the next step, electron beam lithography can be replaced by more standard tools such as project lithography (stepper) for wafer-scale mass production[38]. Although gold is not CMOS compatible, we can use materials such as aluminum[39], or TiN[40] that are CMOS compatible to make nanoantennas. Thus, our design is compatible with standard fabrication processes in IC industry and meet demands for low-cost and mass production.

As the prospects for improving the limit of detection and make a ~cm long NDIR sensor while maintaining the same integration, the two key aspects we are working on are: (1) Reduce the thickness of the LT element from 75 μm to 700 nm (two orders of magnitude)[20]. This can be done by replacing the self-supported LT plate with thin film LT on silicon. (2) Increase the quality factor of the MIM absorber that determines the overlap between the spectral response of the detector and the gas absorption lines. This can be done by optimizing the design of the nanoantennas in the MIM absorbers[30].

Since water vapor (humidity) is a strong source of cross-response in the 5–7 μm, we took several measures to minimize the intereference from the water vapor. First, the NDIR system is installed with a pump that can take out the air and water vapor inside the gas pipelines. Second, we also put desiccants at the outlets of the pipelines to remove water vapor from the ambient. Third, the NDIR system is located on an optical table that is covered by a hood. The hood can help isolate the water vapor

from the human operator. Finally, the measurement lab is well ventilated to minimize the remaining water vapor in the room. Thus in the future, to make sure that the ~cm long NDIR sensors can accurately measure gases in the 5–7 μm region in standard atmospheric conditions, water vapor need to be removed to a very high degree.

Owing to the high insertion loss caused by the gas cell, the current NDIR system is not implemented with a reference channel. In future designs of ~cm long NDIR sensors, one of the detection elements in the 5–7 μm range can be made as a reference detector. Measuring errors caused by dust or diminishing radiation intensity are removed by the use of the reference channel.

In the future, the demonstrated NDIR sensor architecture can be expanded by increasing the number of narrowband detection elements to analyze more target gases (see Supplementary Note 13 for methods of minimizing the thermal cross-talk between the neighboring elements). It can also be extended to other thermal detector platforms such as thermopile detectors and vanadium oxide microbolometers. When combined with the large size focal plane array technology, the presented device architecture will evolve as on-chip infrared spectrometers, which can serve as tools for spectroscopic analysis of gases, chemicals, explosives, and other types of substances.

## Methods

**Finite element simulations**. We used COMSOL, a finite element method-based solver to numerically study the optical properties of the absorbers employing periodic boundary conditions. With plane wave excitation polarized along the y-axis in the electromagnetic waves module. The domain boundaries parallel to the $x-z$ and $y-z$ planes and $port_1$ is the top face of simulation area, $port_2$ is the bottom face of simulation area. Using the corresponding parameter in Supplementary Table 1, we can obtain the optimized absorption curve.

To evaluate the impact of the heat transfer between the sensing area and the environment via convection, we also use the heat transfer module in COMSOL to simulate the temperature change in the detector. The bottom sides of the four silicon posts are all set to be constant temperature $T_0 = 293.15$ K while all other sides of the structure are set as heat insulation with the dimensions of the LT substrate are: length = 2.4 mm, width = 1.5 mm and thickness = 75 μm. The height and radius of the silicon posts are 250 and 50 μm, respectively. As for the MIM absorber, the top layer of nanodisk antennas is ignored for simplicity while the silicon dioxide spacer and gold backplate are included and set as the heat source with a Gaussian profile: $g(x, y) = g_0 * \exp(-x^2/r_0^2) * \exp(-y^2/r_0^2)$. Here $g_0 = P_0/(\pi r_0^2 t)$ is the power density of the heat source; $t = (t_{SiO2} + t_{backplate}) = 0.2$ μm is the total thickness of the spacer and the backplate, the corresponding material parameters are in Supplementary Table 2.

**Device fabrication**. Mid-IR detectors were fabricated at the Center of Micro-Fabrication and Characterization (CMFC) of Wuhan National Research Center for Optoelectronics. A schematic of the fabrication and package of the narrowband detector is shown in Supplementary Fig. 3. The fabrication of the narrowband detectors begins with the deposition of the 80 nm silicon dioxide spacer on top of the gold electrode (100 nm Au/20 nm Cr) pre-deposited on the 75 μm LT substrate (Yamaju Ceramics Co., LTD.) using plasma-enhanced chemical vapor deposition (PECVD). Electron beam resist (AR-P 6200.09) was spin-coated and nanodisk arrays were defined by electron beam lithography (Vistec EBPG 5000plus ES) followed by electron beam evaporation(EB-500S) to deposit 50 nm Au/10 nm Ni (adhesive layer). Another round of electron beam lithography was then performed to define the 600 μm side-length window areas for wire-bonding. The silicon dioxide spacer in the wire-bonding areas was then removed by dry etching (Plasmalab system 100 ICP 180) to expose the gold electrode underneath the spacer. The LT substrate was then cut into separate single-element detectors using a laser cutter, each of which has an antenna array and a wire-bonding area. The detector was then mounted onto a TO-5 packaged impedance matching circuit with a customized special JFET (low drift, low noise) in voltage mode. The bottom electrode of the LT detector is in direct contact with the one input pin of the circuit. The top electrode of the LT detector was then wire-bonded to the other input pin to finish the electrical connection. The optical window is an infrared high transmissive $CaF_2$ glass.

**Measurement of photoresponse**. We use a wavelength tunable quantum cascade laser (QCL) to examine the characteristic of the fabricated detectors. The working wavelength of the QCL (Block Engineering, LaserTune) can be continuously tuned from 5.4 to 6.0 μm. See Supplementary Note 6 for the details about the QCL. An

optical chopper (Thorlabs, MC2000B, $f_{mod} \geq 4$ Hz) was used to mechanically modulate the output beam from the QCL before it reaches the narrowband detectors. The beam modulated by the chopper was then focused by a reflective objective (Thorlabs, LMM-15X-P01) to the area of antenna array. A power supply (GWINSTEK, GPS-3303C) that provides +5 V bias voltage for the impedance matching circuit in the TO-package and a lock-in amplifier (Stanford Research System, SRS-830) connected to a computer controlled by LabVIEW is used to measure the output electrical signal of the detector. The input reference signal of the lock-in amplifier is same as the chopper input signal.

We also build a gas sensing system. A schematic of the gas sensing system is shown in Fig. 5a. The gas sensing system is composed of three subsystems: the optical subsystem contains a collimated SiC broadband IR source (Thorlabs, SLS203L/M, see Supplementary Note 6 for details), an optical chopper, a gas chamber (GAINWAY, GW-1020IR-5M), a reflective objective (Thorlabs, LMM-15X-P01) and the prepared narrowband pyroelectric detector; the electrical subsystem contains a power supply (GWINSTEK, GPS-3303C) that provides +5 V bias voltage for the detector and a lock-in amplifier (SRS-830); the gas supply subsystem contains mass flow controllers (Sevenstar, CS200C) that control real-time flow of each gas. The target gas gets mixed with pure nitrogen gas and sent into the gas chamber.

## Data availability
The data that support the findings of this study are available from the authors on reasonable request; Source data are provided with this paper.

## Code availability
The code that support the findings of this study is available from the authors on reasonable request;

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

## Acknowledgements

F.Y. acknowledges funding support from National Natural Science Foundation of China (NSFC) (11774112, 11604110); National key research and development program of China (2016YFC0201300, 2019YFB2005700); The Fundamental Research Initiative Funds for Huazhong University of Science and Technology (2017KFYXJJ031, 2019kfyRCPY122); Graduates' Innovation Fund, Huazhong University of Science and Technology (5003182041). We thank Li Pan engineer in the Center of Micro-Fabrication and Characterization (CMFC) of WNLO for the support in PECVD fabrication. We thank Zeng Tiantian engineer in the Huazhong University of Science & Technology Analytical & Testing Center for the support in FTIR test. We thank the technical support from Experiment Center for Advanced Manufacturing and Technology in School of Mechanical Science & Engineering of HUST.

## Author contributions

X.C.T. and F.Y conceived the idea. X.C.T. and J.Y.L. designed and simulated the prototype structures. X.C.T. fabricated the detector and built up the test system. X.C.T. and H.Z. performed the photoresponse and gas sensing measurement. X.C.T., H.Z., and H.W.W. performed the electrical calculation, data analysis. Q.S.G. helped with data analysis. H.B.Z. provided the Lithium Tantalate materials. F.Y. organized the project, analyzed the results, and provided the support. All authors contributed to the preparation of the manuscript.

## Competing interests

The authors declare no competing interests.
