## [Peer Review File · Nature Communications]

Reviewers' comments:

Reviewer #1 (Remarks to the Author):

The authors describe a plasmonic-pyroelectric detector arrays as a compact non-dispersive infrared sensing of specific gas molecules. The idea itself is intriguing but there are major issues that should be addressed before the manuscript is suitable for publication in Nature Communications. I really try not to be overly picky when reviewing manuscripts, but several issues stood out that I feel really need to be addressed, and it is for these reasons I cannot recommend it for publication:

--The authors fabricated a pyroelectric detector using lithium tantalite as the active layer. The Reviewer is familiar with the pyroelectric perovskite lithium tantalate but was unable to find references to lithium tantalite. The detector structure itself is very timely; pyroelectric-plasmonic detectors are becoming a hot research area, and the structure itself is appropriate for the journal.

--Pyroelectric detectors are not typically biased. There are legitimate reasons why one would be biased, such as J. V. Mantese's active mode detection (Appl. Phys. Lett., 90, 113503 (2007)), but since these detectors use a mechanical chopper, the reason for a +5V DC bias is unclear.

--The noise equivalent power for any detector should equal the square root of the quadratic summation of each individual noise contribution: $NEP = \sqrt{J^2 + T^2 + \dots}$, where J is Johnson noise and T is temperature noise. Based on this, the summation of noise sources at any frequency in Figure 4C should equal the corresponding point in Figure 4D, but that is not the case. Noise measurements are inherently extremely difficult, so this is a completely understandable issue.

--The sensor was designed to simultaneously measure the concentrations of eight analyte gasses, yet the paper only describes sensing carbon monoxide and sulfur dioxide mixtures. The authors successfully tested each of the eight gasses separately, so it is unclear why measurements of all eight at once were not presented, since that was the original purpose of the paper.

Reviewer #2 (Remarks to the Author):

This paper reports the development of a multi-wavelength filter-free low-cost mid-infrared detector. The potential of this source for biomedical and industrial applications is then demonstrated by applying it to simultaneous multi-gas detection.

This is a timely and highly relevant piece of research for the advancement of mid-infrared integration technologies, where numerous possibilities exist for sensors integrated into smart phones, watches, or wearables, and I am pleased to recommend this paper for publication in Nature Communications.

My other comments are given below:

1. Can the authors comment on the CMOS compatibility of their design? This would be a desirable criterion to meet demands for, e.g., low-cost and mass production.

2. I am not sure how much Refs 1-8 given in the first sentence discuss "highly characteristic rotational or vibrational transition bands". I suggest the authors check these carefully.

3. The novelty here is the NDIR sensor and especially the simultaneous multi-gas detection, which is a very significant result in per se. However, plasmonic-enabled multi-wavelength detector arrays are not a novel concept, and I would encourage the authors to compare their technology with current work, e.g., [Opt. Mater. Express 8, 1696 (2018); Optica 4, 276 (2017), ACS Photonics 4, 1371-1380 (2017), Sci. Rep. 5, 17451 (2015)], or work done by the same authors [SPIE 105361H (2018), <https://doi.org/10.1117/12.2286437>].

4. Some sections of the paper seem a bit disconnected. E.g., is not clear how "Heat generation in the detector" is relevant to the main results, and as a reader I am not quite sure what I need to look for in this

section, etc. Same applies to, e.g., "Photoresponse of the narrowband detectors". Although some numbers are provided it is not clear how the performance compares with current state-of-the-art detectors, and, e.g., how is this relevant to the main results claimed here?

5. Can the authors provide information on the optical power per wavelength used?

6. The spectral profiles in Figure 2 do not cover the full wavelength range, and most of them seem to be randomly truncated. I suspect this was done to hide out of band absorption (plasmonic structures are known to leak in these applications). Can the authors comment on this?

The formatting of this response is to match that of the reviewers. Each of the valuable critique points are addressed directly, hopefully as clearly as possible.

In response to Reviewer 1:

The authors describe a plasmonic-pyroelectric detector arrays as a compact non-dispersive infrared sensing of specific gas molecules. The idea itself is intriguing but there are major issues that should be addressed before the manuscript is suitable for publication in Nature Communications. I really try not to be overly picky when reviewing manuscripts, but several issues stood out that I feel really need to be addressed, and it is for these reasons I cannot recommend it for publication:

--The authors fabricated a pyroelectric detector using lithium tantalite as the active layer. The Reviewer is familiar with the pyroelectric perovskite lithium tantalate but was unable to find references to lithium tantalite. The detector structure itself is very timely; pyroelectric-plasmonic detectors are becoming a hot research area, and the structure itself is appropriate for the journal.

Lithium tantalite(LT) is one of a large number of octahedral ferroelectrics. Its structure is similar to lithium niobate (LN). The following literatures describe LT and related pyroelectric detectors:

1. Batra A-K, Aggarwal M-D. "Chapter 4 Important Pyroelectrics: Properties and Performance Parameters," in *Pyroelectric Materials: Infrared Detectors, Particle Accelerators, and Energy Harvesters* (SPIE Press, 2013);
2. Chatard J-P, Norkus V, Dennis PNJ. Pyroelectric infrared detectors based on lithium tantalate: state of art and prospects. 5251, 121 (2004).
3. Roundy C-B. "Pyroelectric self-scanning infrared detector arrays," *Applied Optics* 18, 943-945 (1979).
4. Stenger V, Shnider M, Sriram S, Dooley D, and Stout M. "Thin Film Lithium Tantalate (TFLT) pyroelectric detectors", *Proc. SPIE 8261, Terahertz Technology and Applications V*, 82610Q (22 February 2012);

We choose LT as the active layer because it is a very mature pyroelectric material that is readily available commercially. It can be purchased from the following vendors in the form of both substrates or thin films and the available wafer sizes are 3, 4 or 6 inches:

1. <http://www.yamajuceramics.co.jp/english/products/>
2. <https://www.nanoln.com/>

--Pyroelectric detectors are not typically biased. There are legitimate reasons why one would be biased, such as J. V. Mantese's active mode detection (*Appl. Phys. Lett.*, **90, 113503 (2007), but since these detectors use a mechanical chopper, the reason for a +5V DC bias is unclear.**

We thank the reviewer for raising this point.

The **LT element** is mounted on a TO-5 packaged impedance matching circuit with a customized special JFET (low drift, low noise) in voltage mode. The +5V DC bias voltage from the power supply is actually applied on the **impedance matching circuit in the TO-package**, rather than the LT element. To clarify this point, we have updated the "Sample fabrication" and "Measurement of photoresponse" sections in the main text accordingly.

--The noise equivalent power for any detector should equal the square root of the quadratic summation of each individual noise contribution: $NEP = \sqrt{J^2 + T^2 + \dots}$, where J is Johnson noise and T is temperature noise. Based on this, the summation of noise sources at any frequency in Figure 4C should equal the corresponding point in Figure 4D, but that is not the case. Noise measurements are inherently extremely difficult, so this is a completely understandable issue.

In Figure 4C, the purple line with triangles is the fluctuation (defined as the “total noise”) in the output voltage of the packaged detector without input optical power, measured by a lock-in amplifier (SRS-830). While the red line with squares and the black line with filled circles represent the calculated values of Johnson noise and temperature noise, respectively.

There exists a discrepancy between the measured “total noise” (purple) and the square root of the quadratic summation of the Johnson noise and the Temperature Noise: $\sqrt{J^2 + T^2}$. The discrepancy is mainly attributed to the fact that the packaged detector is not installed in a vacuum chamber. Besides, the TO-5 package of the detector is neither sealed by a bandpass filter nor by an IR transmissive window, so the LT element is directly exposed to atmosphere. Any ambient temperature fluctuation during the measurement could contribute to the fluctuation in the output voltage of the detector without input optical power. Besides, the low frequency noise of the readout circuit could be another factor of the fluctuation in the output voltage.

As pointed out by the reviewer, noise measurements are inherently extremely difficult. We believe that when the measurement condition is fully optimized, the measured “total noise” should approach the calculated value of $\sqrt{J^2 + T^2}$ that represents the lower limit of the detector noise.

--The sensor was designed to simultaneously measure the concentrations of eight analyte gasses, yet the paper only describes sensing carbon monoxide and sulfur dioxide mixtures. The authors successfully tested each of the eight gasses separately, so it is unclear why measurements of all eight at once were not presented, since that was the original purpose of the paper.

During the preparation of the manuscript, we only had three mass flow controllers, so we were only able to mix two target gases and the third mass flow controller is for nitrogen gas. That is why the submitted manuscript only describes sensing carbon monoxide and sulfur dioxide mixtures.

To respond to this question, we purchased more mass flow controllers and upgraded the gas supply subsystem. We then conducted the NDIR experiment on the mixture of eight target gases using five narrowband detectors. The performances of the remaining three narrowband detectors were found to be compromised and this is possibly due to improper storage.

The NDIR experiment on the mixture of eight gases is very time consuming. Due to the outbreak of the 2019-Wuhan-novel coronavirus (COVID-19), the lab building is shut down and we are unable to fabricate new detectors. So in the revised Supplementary Note 8 we provided the results of the NDIR experiment on the mixture of eight target gases using five narrowband detectors. Hopefully these data will strengthen the conclusion of this paper.

In response to Reviewer 2:

This paper reports the development of a multi-wavelength filter-free low-cost mid-infrared detector. The potential of this source for biomedical and industrial applications is then demonstrated by applying it to simultaneous multi-gas detection.

This is a timely and highly relevant piece of research for the advancement of mid-infrared integration technologies, where numerous possibilities exist for sensors integrated into smart phones, watches, or wearables, and I am pleased to recommend this paper for publication in Nature Communications.

My other comments are given below:

1. Can the authors comment on the CMOS compatibility of their design? This would be a desirable criterion to meet demands for, e.g., low-cost and mass production.

First, lithium tantalate(LT) is a very mature pyroelectric material that commercially available in the form of both free-standing ultra-thin LT wafers or thin film LT on silicon wafers. The diameter of the free-standing LT wafers can be as large as 4 inches and the thickness of the wafer can be as thin as 10 μ m. The diameter of the thin film LT on silicon wafers can be as large as 6 inches and the thickness of LT thin film can be as thin as 300 nm. (Please refer to <https://www.nanoln.com/> for details about the available LT products). Pyroelectric detectors based on LT have been widely used in applications such as motion detection, non-contact temperature measurement, NDIR gas analysis, and IR spectrometers. These detectors were fabricated and packaged using the standard tools for making IC chips. Therefore we choose LT to be the active layer of our design.

Second, the nanoantenna array was also fabricated using regular fabrication processes in IC industry such as electron beam lithography, electron beam evaporation, and metal lift-off. In the next step, electron beam lithography can be replaced by more standard tools such as project lithography (stepper) for wafer-scale mass production. Although gold is not CMOS compatible, we can use materials such as aluminium [Acs Nano 8, 834-840 (2014)], or TiN [Adv Mater 26, 7959-7965 (2014)] that are CMOS compatible to make nanoantennas.

Our design is based on the low-cost LT pyroelectric detectors and further replaces the separate bandpass filters with on-chip absorption filters (the nanoantenna array). When successfully mass-produced, the cost of patterning metal nanostructures should be lower than making separate bandpass filters.

To conclude, our design is compatible with standard fabrication processes in IC industry and meet demands for low-cost and mass production.

2. I am not sure how much Refs 1-8 given in the first sentence discuss “highly characteristic rotational or vibrational transition bands”. I suggest the authors check these carefully.

We thank the reviewer for catching this. We have updated the references given in the first sentence using the following literature:

1. Popa D, Udrea F. Towards Integrated Mid-Infrared Gas Sensors. Sensors (Basel,

Switzerland) 19, 2076 (2019).

2. Gordon IE, et al. The HITRAN2016 molecular spectroscopic database. *Journal of Quantitative Spectroscopy and Radiative Transfer* 203, 3-69 (2017).

3. Mantsch HH, Chapman D. *Infrared spectroscopy of biomolecules*. Wiley-Liss New York (1996).

4. Barth A. *Infrared spectroscopy of proteins*. *Biochimica et Biophysica Acta (BBA) - Bioenergetics* 1767, 1073-1101 (2007).

5. Schliesser A, Picqué N, Hänsch TWJNP. Mid-infrared frequency combs. *Nature Photonics* 6, 440 (2012).

3. The novelty here is the NDIR sensor and especially the simultaneous multigas detection, which is a very significant result in per se. However, plasmonic enabled multi-wavelength detector arrays are not a novel concept, and I would encourage the authors to compare their technology with current work, e.g., [Opt. Mater. Express 8, 1696 (2018); Optica 4, 276 (2017), ACS Photonics 4, 1371-1380 (2017), Sci. Rep. 5, 17451 (2015)], or work done by the same authors [SPIE 105361H (2018), <https://doi.org/10.1117/12.2286437>].

We agree with the reviewer that using plasmonic nanostructures to tailor the responses of infrared detector pixels and construct multi-wavelength detector array is not a novel concept. Here are some typical examples:

Since 2012, Shinpei Ogawa et al. reported using plasmonic metamaterial absorbers to tailor the response of thermopile detectors and construct wavelength selective and polarization selective detector arrays [*Applied Physics Letters*, 2012,100 (2):021111; *Optical Engineering*, 2013,52 (12):127104; *Sensors*,2015,15 (6):13660-13669; *Materials*, 2017,10 (5):493].

Since 2013, we also reported using nanoantenna array to tailor the response of nanomechanical detectors [*Nano Letters*, 2013,13 (4):1638-1643; *IEEE Photonics Technology Letters*, 2014, 26 (2):202-205; *Plasmonics* (2017) 12: 1921].

In 2017, Willie Padilla et al. reported using plasmonic metamaterial absorbers to construct narrowband pyroelectric detectors based on thin film lithium niobate (TFLN) [*Optica* 4, 276 (2017), as listed by the reviewer].

However, the question that remains is: **what practical applications can these narrowband detectors deliver?** ["Focusing in on applications." *Nature Nanotech.* 10, 1 (2015) doi:10.1038/nnano.2014.332] Therefore we choose NDIR multigas sensing as a real application to examine the usefulness of these new detectors.

1. [CMOS-compatible mid-IR metamaterial absorbers for out-of-band suppression in optical MEMS, *Opt. Mater. Express* 8, 1696 (2018)]: This paper reports a metal-insulator-metal(MIM) absorber fabricated using masked UV (i-line) lithography. To make the fabrication more CMOS-compatible, aluminium is used for the top and the bottom layer. The structure demonstrated in this paper is not a narrowband detector but a narrowband absorber. But it is still very useful to us because we also plan to replace the currently used electron beam lithography with masked UV lithography and try wafer-scale production on 4 inch LT substrate. Aluminium is also a good alternative to gold due to its low cost and CMOS compatibility.

2. [Multifunctional metamaterial pyroelectric infrared detectors, *Optica* 4, 276 (2017)]: This paper reports thin film lithium niobate (TFLN) based pyroelectric narrowband detectors enabled by MIM absorbers in which the 575 nm thick TFLN serves as the dielectric spacer. Narrowband (560 nm FWHM at 10.73 μm) detection with a thermal time constant of 28.9 ms with a room temperature detectivity, D^* , of $10^7 \text{ cm W} / \text{Hz}^{1/2}$ is achieved. This is definitely a pioneering work and also very useful to us. The use of thin film LN instead of bulk LN as the active layer significantly reduce the total thickness of the detector, which in turn leads to higher temperature increase and detector response. LT has a similar structure as LN but its pyroelectric coefficient is slightly higher. Thin film lithium tantalite (TFLT, thickness range: 300 nm to 900 nm) on 6 inch silicon wafers is readily available commercially (Please refer to <https://www.nanoln.com/> for details about the available LT products). Therefore we also plan to try metamaterial enabled narrowband detectors using TFLT as the active layer and examine their performances in NDIR multigas sensing.

3. [On-Chip Narrowband Thermal Emitter for Mid-IR Optical Gas Sensing, *ACS Photonics* 4, 1371-1380 (2017)]: This paper reports an on-chip narrowband thermal light source realized by heating an MIM structure enclosed by an Al_2O_3 sealing layer. Narrowband emission at the center wavelength of 3.96 μm with a resonance quality factor of 15.7 and an emissivity of 0.99 is demonstrated. A filter-free NDIR gas sensing setup is constructed to perform single target gas sensing on CO_2 using the developed narrowband thermal light source. The target of this work is the same as our work: to realize a compact filter-free NDIR gas sensor. But the difference is that this work uses the heated MIM structure as the narrowband light source of the NDIR sensor while our work uses the MIM structure to tailor the responses of the detector in the NDIR sensor. Besides, this work does not demonstrate NDIR multigas sensing while our work does.

4. [A highly efficient CMOS nanoplasmonic crystal enhanced slow-wave thermal emitter improves infrared gas-sensing devices, *Sci. Rep.* 5, 17451 (2015)]: This paper reports a tungsten-based narrowband thermal emitter by tailoring thermal plasmon emission using the slow-wave lattice resonance in a designed plasmonic crystal. The structure of the narrowband emitter is “tungsten nanoantenna array – silicon dioxide spacer – tungsten backplate” enclosed by silicon dioxide and silicon nitride passivation layer. The developed narrowband emitter is then inserted in a conventional NDIR gas sensor (with bandpass filter) for single target gas (CO_2) sensing. The result shows that the emission intensity of the engineered narrowband emitter is enhanced by almost 4-fold compared to a standard non-plasmonic emitter. The idea of work is similar to [ACS Photonics 4, 1371-1380 (2017)] by using plasmonic metamaterial to tailor the emission of the infrared source. But the developed narrowband IR emitters do not eliminate the bandpass filters in the NDIR sensor. Besides, this work does not demonstrate NDIR multigas sensing either.

5. [Narrowband plasmonic metamaterial absorber integrated pyroelectric detectors towards infrared gas sensing, *SPIE* 105361H (2018), <https://doi.org/10.1117/12.2286437>]: This is a conference paper we presented at SPIE OPTO in Feb 2018. In this paper, we proposed the idea of using narrowband plasmonic metamaterial absorber integrated pyroelectric detectors to replace the “bandpass filter + detector” pairs in conventional NDIR gas sensor. We also reported the preliminary results of 8 MIM absorbers that correspond to 8 target gases. But at that time we had not finished the fabrication of the narrowband pyroelectric detectors, so this conference paper does not contain the results of the detector responses and the NDIR multigas sensing.

3. Some sections of the paper seem a bit disconnected. E.g., is not clear how “Heat

generation in the detector” is relevant to the main results, and as a reader I am not quite sure what I need to look for in this section, etc. Same applies to, e.g., “Photoresponse of the narrowband detectors”. Although some numbers are provided it is not clear how the performance compares with current state-of-the-art detectors, and, e.g., how is this relevant to the main results claimed here?

We thank the reviewer for the valuable comments. In this work, we want to report two main achievements:

1. We accomplished a general purpose narrowband pyroelectric detector architecture by integrating MIM absorbers with LT active layer. To our best knowledge, this is the first demonstration of such a combination. Although similar idea has been reported by Willie Padilla et al. [Optica 4, 276 (2017)], the active material in their paper is LN rather than LT.
2. We demonstrated NDIR multigas sensing using the developed narrowband pyroelectric detectors.

As a general purpose detector architecture, we feel that it is necessary to first provide a full description about the working principles of the detectors before reporting the results of NDIR multigas sensing. The section “Heat generation in the detector” describes how the heat generated by the dissipated optical power leads to the temperature increase in the detector. The section “Photoresponse of the narrowband detectors” describes the corresponding detector responses and noises.

As suggested by the reviewer, we added Supplementary Note 11 the comparison between the performance of our detectors and the commercial single element LT detectors reported in the following papers:

Chatard J-P, Norkus V, Dennis PNJ. Pyroelectric infrared detectors based on lithium tantalate: state of art and prospects. In: Detectors and Associated Signal Processing) (2004).

From the comparison it is seen that the narrowband detectors enabled by plasmonic metamaterial absorber is comparable to the commercial single element LT detectors that use metal black coating as the IR absorber in terms of the specific detectivity. Since the LT element in this work is relatively thick (75 μm) and the narrowband detectors are not vacuum packaged, it is expected that the specific detectivity can be further improved by using thinner LT elements and implementing vacuum package.

For more data about the performance of commercial single element LT detectors, please refer to:

<https://www.dias-infrared.com/products/infrared-detectors-and-arrays/pyroelectrical-detectors-pyrosens>

5. Can the authors provide information on the optical power per wavelength used?

In the revised Supplementary Note 9, we provided the output power of the two light sources as a function of wavelength:

- a) Global IR source SLS203L/M from Thorlabs

https://www.thorlabs.com/newgrouppage9.cfm?objectgroup_id=7269&pn=SLS203L/M

b) Widely Tunable Mid-Infrared Quantum Cascade Laser LaserTune™ from Block Engineering

<https://www.blockeng.com/products/lasertune.html>

6. The spectral profiles in Figure 2 do not cover the full wavelength range, and most of them seem to be randomly truncated. I suspect this was done to hide out of band absorption (plasmonic structures are known to leak in these applications). Can the authors comment on this?

The measured absorption spectra in the full wavelength range of the FTIR are provided in the Supplementary Note 10. It is true that for each absorber, there are out-of-band absorption peaks that are weaker than the main absorption peak in other wavelengths. The out-of-band absorption in the absorber will cause the detector to respond to other out-of-band gases and this is the cross-talk problem. Our approach to solve the cross-talk problem is: if there are M target gases in the chamber, one can implement N ($N \geq M$) narrowband detectors with different detection wavelengths to measure the gas mixture. The concentration of each target gas in the mixture can then be back calculated from the responses of the M detectors.

REVIEWER COMMENTS

Reviewer #2 (Remarks to the Author):

The authors addressed some of the comments. However, a few things remain unclear:

3. Authors: "However, the question that remains is: what practical applications can these narrowband detectors deliver? ["Focusing in on applications." Nature Nanotech. 10, 1 (2015) doi:10.1038/nnano.2014.332] Therefore we choose NDIR multigas sensing as a real application to examine the usefulness of these new detectors."

From a real application perspective, the main challenge in the mid-IR, and the core of a spectrometer in general (NDIR included) is the light source and not the detector. Commercial NDIR sensors, which can cost a few dollars (e.g. Sensirion) use MEMS heaters which generate a fraction of the power available from the IR source used here. In addition, they are ~cm long (some <1cm), and not 5m as in this work. Clearly the sensitivity of an NDIR will depend (strongly) on these two parameters, probably the reason why the authors used such a bulky system, but I can't possibly see how this system can be considered a real application, assuming, of course, its cost is comparable to what is commercially available, a comment the authors were somewhat evasive about.

5. The authors did not answer this question. Providing the datasheet for the two light sources does not answer the question. This is very important (as explained above) as it would give a sense of how this technology compares with available NDIR sensors. For example, the authors present the detection limit of their sensor, but this clearly depends on the amount of power they used in their experiment, and I am not even mentioning the 5m long sensor...

It would be nice if the authors addressed these points as it would be misleading to the reader, i.e. led to believe the manuscript presents a full NDIR sensor (as claimed in the abstract) better than what is currently available, which clearly is untrue. In reality the NDIR presented here is significantly bulkier and probably way more expensive than most current NDIRs. The authors must be clear about what is the exact novelty here, i.e. multi-gas sensing with the plasmonic detector only (not even the detector itself).

Reviewer #3 (Remarks to the Author):

The paper presents a novel integrated array of thermal photodetectors, independently addressed, each using a different geometry (size and lattice constant) of nanostructured antennae. The photodetectors thereby become narrowband absorbers suitable for use in gas detection via nondispersive infrared sensing. The level of integration achieved using this technique is a step forward, novel and interesting. I have no doubt that the device construction and attention to detail in its modelling is fine work, and it is an achievement for all the antennae on one device to have their absorption spectra line up so nicely with all the identified gases. This means the optical modelling and level of fidelity achieved during fabrication must have been of high quality.

However, I do have some scientific reservations about the paper, as currently presented. These mainly relate to the instrumentation aspects of the work, i.e. to its design and use as a gas sensor, the experiments performed, and the analysis of the results. On this basis I would conclude that the paper needs a serious rewrite before it is ready for publication in a journal such as this. Given that the authors may currently be unable to access their lab, some of the claims in the paper may have to remain unsubstantiated (for example the measurement of the 3dB point in the frequency response) and the discussion / conclusions should avoid overstepping their experimental basis.

Scientific

1. A comment on the first reviewer's point about lithium tantalite. As a reminder, this was the discussion between reviewer and authors: [Reviewer] The authors fabricated a pyroelectric detector using lithium tantalite as the active layer. The Reviewer is familiar with the pyroelectric perovskite lithium tantalate but was unable to find references to lithium tantalite. The detector structure itself is very timely; pyroelectric-plasmonic detectors are becoming a hot research area, and the structure itself is appropriate for the journal. [Authors] Lithium tantalite(LT) is one of a large number of octahedral ferroelectrics. Its structure is similar to lithium niobate (LN). I note that the supplementary material includes reference to the material LiTaO₃, which is conventionally referred to as lithium tantalate. Therefore I assume that there is either a typo in the main text (it should be lithium tantalate) or in the equation in the supplementary data. The supporting references provided by the authors show that we might be talking about lithium tantalate all along.
2. Page 3: lithium tantalate (LT) is described as a good IR absorber, but the effect of occluding this with a gold electrode has not been discussed. Later it is stated that the nanodisk antennae on top convert the IR radiation to heat and the LT material is described as being a heat absorber (presumably via conduction not radiation) therefore the IR absorbing properties of LT are irrelevant. This section therefore needs to be rewritten because at present it is inconsistent. The reason for the choice of LT as sensor material needs to be made clear. Surely the main criterion should be that the material has a high conversion of heat to electrical current.
3. The absorption spectra show nice targeting of the different gases under consideration. One gas I would especially expect a strong cross-response to is water vapor (humidity) in the 5-7 μ m region, which will make the measurement of formaldehyde and NO₂ especially challenging. As human experimenters are a major source of variable concentrations of water vapor in the lab, this will also make measurement of sensor performance challenging as well.
4. An additional comment, worthy of note in the early sections on design and in the discussion of results, is this. I am not sure why the authors omitted a reference photodetector, as typically used in gas sensing via NDIR. Typically, a narrowband detector centered on 3.90 μ m or 3.95 μ m is used for this purpose, as it shows minimal cross-response to many gases. It accounts for broad spectrum variability in the light source intensity. Without it, device performance will be affected.
5. The difference between the calculated and measured voltage response is over an order of magnitude. The authors propose that this is a result of non-vacuum packaging; presumably then, over 90% of the device heat is lost to convection within a timescale more rapid than the modulation frequency of 4Hz. 90% sounds rather large, especially as convection within a small cell will be hampered. Some additional evidence would be needed to substantiate this claim.
6. The 3dB frequency is measured relative to 4Hz, which is the lowest frequency supported by the chopper rather than the frequency corresponding to peak response. Inspection of supplementary Fig 5 confirms this. This makes the measured 3dB point arbitrary, or irrelevant. The experiment on frequency response therefore can't be used in support of any particular 3dB point. To measure this properly, a chopper would be needed that supported lower frequency operation, going sufficiently low to cover the frequency of peak response. They are commercially available, but I assume the authors don't have one.
7. I don't understand why the calculated Johnson noise isn't white. The calculation isn't explained – which resistor is generating the Johnson noise? Most typically, the limiting Johnson noise will be produced by the large transimpedance in a current-to-voltage amplifier, if that is what is being used. Without a circuit diagram for the first stage amplification it is impossible to comment. A circuit diagram (actual or equivalent major

elements) should be provided. The authors must have some idea of what it is (even though they use a commercial amplifier) since they have been able to calculate the Johnson noise themselves.

8. On the use of the modified Beer Lambert Law, equation (1) I don't understand the reasoning behind accounting for diffraction of the light beam using parameter c , since this should be unaffected by the presence of gas (the gases have a relatively small contribution to the refractive index of the air and therefore changes to beam width with concentration should be negligible). I am uncertain as to where the authors got equation (1) from. It is not in their reference [31] (Barnes JR, et al. A Femtojoule Calorimeter Using Micromechanical Sensors. Rev Sci Instrum 65, 3793-3798 (1994)) nor in the associated erratum for that paper. 5m would be a very long gas cell to have in the lab, was this a multipass cell? It should be described. I suspect it may be a typo. Fig 5(a) does not show a collection lens to collimate the beam from the globalar before passing through the cell. I would struggle to measure the intensity of a globalar with a thermal detector of the small size indicated here, at a distance of 5m.

The value of κ required in equation (1) is a composite: the integration of κ values for individual absorption lines over the spectral region that is covered by the detector. The integration introduces an additional nonlinearity, over and above that of the Beer Lambert law, since each of the individual gas lines obeys its own Beer-Lambert equation and becomes nonlinear at a different concentration. This needs to be accounted for as it is important for the later comparison of experiment with theory. The concentrations used are relatively high, the pathlength is long and the results themselves indicate that measurements were made in the nonlinear regime of the Beer-Lambert law where this additional nonlinearity becomes significant.

The authors should modify equation (1) accordingly, and the analysis of data using this equation. Either that or the authors must provide a robust scientific justification for the use of this equation, along with the value of c that applies to their instrument.

9. The mixed gas experiment is welcome, but confusing. The voltages should all be referenced to zero conditions, ie V_0 should be measured in nitrogen and not with the fixed gas concentration. I'm not at all sure whether this has been done. Fig 6(a) shows the level of cross-response that one might expect from the original plots of absorbance of the MIMs. Fig 6(b) shows a much more ambiguous response. If one were given the voltage on each channel, would one be able to work out the concentrations of each of the gases? It's not clear from the data in Fig 6(b). That is why I wonder whether the voltages were not referenced to zero gas.

For this section to be convincing, the authors should turn the problem around and see whether their model allows them to accurately work out the concentrations from the detected voltages (and test this on the data).

10. Reviewer 1 commented that not enough gas testing had been completed. In my view it is not necessary to perform measurements on all gases at once, to characterize the cross-response pairwise across the range would be adequate, limiting the measurements to those gases that are neighbors on the spectrum or where MIMs are physical neighbors, to show a worst case.

11. The question of whether there is thermal cross-talk between the MIMs remains unanswered, but is important for device integration, having been previously shown by others (using different technology) to be a limiting concern. Is the cross-talk entirely predicted by the optical overlap between nanodisk absorption spectra, or is there additional cross-talk to be accounted for?

12. Using their knowledge of noise and gas response, the authors should estimate a limit of detection for each gas and the expected level of cross-response for each of the neighboring gases on the spectrum. Limits of

detection for some commonly measured gases (eg CO₂ and possibly CO, CH₄) should be provided in the abstract to help others prioritize the paper.

The case for “accurate detection” as mentioned in the abstract is not made, and anyway should be quantified: what level of accuracy?

13. I generally agree with the discussion of the device’s potential in imaging and on-chip spectroscopy. However, its performance for NDIR sensing is not yet evidenced. The authors have not estimated a limit of detection, but I would be surprised if this was comparable to conventional devices. The main limitation seems to be the small absorber area, which is a natural consequence of the high level of integration and desire to contain all this sensing capability within a small device. The authors should therefore provide additional comment on the prospects for improving the limit of detection while maintaining this integration, if they want this to be a potential application.

Presentation

14. Page 2. The performance of NDIR sensors is not “unparalleled”. Sensors based on tunable diode laser spectroscopy offer better sensitivity and selectivity. The reason why NDIR sensors are so important is (a) their low cost compared to such alternatives, and (b) the fact that other, potentially lower cost sensors are ineffective for certain gases such as CO₂.

15. I could not open the supplementary dataset. It is conventional for open data to use commonly accessible and stable file formats such as csv.

16. Page 2 “Moreover, this scheme does not allow the real-time monitoring of multiple gases at the same time.” I disagree; it would be perfectly possible to have such multiple filter / detector sensors operating simultaneously if the detector elements are independently addressed.

In response to Reviewer 2:

The authors addressed some of the comments. However, a few things remain unclear:

3. Authors: "However, the question that remains is: what practical applications can these narrowband detectors deliver? ["Focusing in on applications." Nature Nanotech. 10, 1 (2015) doi:10.1038/nnano.2014.332] Therefore we choose NDIR multigas sensing as a real application to examine the usefulness of these new detectors."

From a real application perspective, the main challenge in the mid-IR, and the core of a spectrometer in general (NDIR included) is the light source and not the detector.

Commercial NDIR sensors, which can cost a few dollars (e.g. Sensirion) use MEMS heaters which generate a fraction of the power available from the IR source used here. In addition, they are ~cm long (some <1cm), and not 5m as in this work. Clearly the sensitivity of an NDIR will depend (strongly) on these two parameters, probably the reason why the authors used such a bulky system, but I can't possibly see how this system can be considered a real application, assuming, of course, its cost is comparable to what is commercially available, a comment the authors were somewhat evasive about.

To clarify, the gas cell used in this work is a White type multipass cell with an effective optical length of 5m. The physical length and width of the gas cell are 448mm and 72mm, respectively. The detailed optical configuration of the White multipass cell is presented in Supplementary Figure S10(b).

We agree with the reviewer that the NDIR system in its current form is still bulkier than the commercial NDIR sensors. As shown by the Supplementary Figure S10 in Supplementary Note 9, it is now designed as a versatile laboratory setup to characterize the photoresponses of the developed narrowband detectors and test them in the NDIR multigas experiment with as many target gases as possible.

To make a ~cm long NDIR sensor, there are two things we working on:

- 1) Reduce the thickness of the LT element from 75 μ m to 700 nm (two orders of magnitude). This can be done by replacing the self-supported LT plate with thin film LT on silicon. The corresponding pyroelectric current i_p and specific detectivity D^* will increase by two orders of magnitude.
- 2) Increase the quality factor Q of the MIM absorber to >30 . For example, when $Q = 30$, the FWHM of the MIM absorber with a peak wavelength of 4.26 μ m (for CO₂) is 142 nm, the effective absorption coefficient of CO₂ gas is calculated to be $k_{\text{eff}}=7.1 \times 10^{-6}$, and the required optical path length can be reduced to 30 cm. By using a multi-pass design, the physical length of the sensor should be able to achieve sub-1 cm range.

5. The authors did not answer this question. Providing the datasheet for the two light sources does not answer the question. This is very important (as explained above) as it would give a sense of how this technology compares with available NDIR sensors. For example, the authors present the detection limit of their sensor, but this clearly depends on the amount of power they used in their experiment, and I am not even mentioning the 5m long sensor...

The original question of the reviewer is “Can the authors provide information on the optical power per wavelength used?”

Here we assume that the reviewer is asking for the optical power that arrives at the narrowband detectors, rather than the direct output of the light sources.

To find out the the optical power per wavelength (spectral density) used, we use the narrowband detector shown in Fig. 4a in the maintext as a specific example. The spectral absorption of the built-in absorber has a peak absorption wavelength $\lambda_{\text{peak}} = 5.52 \mu\text{m}$ with a FWHM = 670 nm. The absorption rate at λ_{peak} is $A(\lambda_{\text{peak}}) = 0.8$.

Assuming the spectral density of the optical power arriving at the detector at λ_{peak} is $P_{\text{arrive}}(\lambda_{\text{peak}})$, and the relative spectral lineshape of the broadband IR source is $\eta(\lambda)$, where $\eta(\lambda)$ is normalized to the output spectral power density at λ_{peak} , then the spectral density of the optical power arriving at the detector can be described as $P_{\text{arrive}}(\lambda) = P_{\text{arrive}}(\lambda_{\text{peak}}) \cdot \eta(\lambda)$.

The spectral density of the optical power absorbed by the detector is then described as $P_{\text{absorbed}}(\lambda) = P_{\text{arrive}}(\lambda_{\text{peak}}) \cdot \eta(\lambda) \cdot A(\lambda)$, where $A(\lambda)$ is the spectral absorption of the built-in absorber. The total optical power absorbed by the detector P_{absorbed} is:

$$P_{\text{absorbed}} = \int P_{\text{arrive}}(\lambda_{\text{peak}}) \cdot \eta(\lambda) \cdot A(\lambda) \cdot d\lambda = P_{\text{arrive}}(\lambda_{\text{peak}}) \cdot \int \eta(\lambda) \cdot A(\lambda) \cdot d\lambda$$

and $P_{\text{arrive}}(\lambda_{\text{peak}})$ can be calculated as:

$$P_{\text{arrive}}(\lambda_{\text{peak}}) = P_{\text{absorbed}} / \int \eta(\lambda) \cdot A(\lambda) \cdot d\lambda$$

The voltage response of the detectors $R_{\text{detector}} = V_{\text{out}} / P_{\text{QCL}}$ is characterized by a tunable QCL with its output wavelength tuned to λ_{peak} and the measured voltage response $R_{\text{detector}} = 90 \text{ V/W}$. It means that if the output voltage of the detector is $V_{\text{out}} = 2.4 \text{ mV}$, the total absorbed optical power P_{absorbed} is found to be $P_{\text{absorbed}} = V_{\text{out}} / R_{\text{detector}} \cdot A(\lambda_{\text{peak}}) = 2.4 \times 10^{-3} \text{ V} / 90 \text{ (V/W)} \cdot 0.8 = 4.4 \times 10^{-6} \text{ W} = 21.3 \mu\text{W}$.

In the NDIR experiment with the broadband global IR source (SLS203L/M, Tholabs), the typical measured output voltage of the detectors is $V_{\text{out}} = 2.4 \text{ mV}$ and the corresponding total absorbed optical power P_{absorbed} is $21.3 \mu\text{W}$. From the above calculation, we have

$$P_{\text{arrive}}(\lambda_{\text{peak}}) = 21.3 \mu\text{W} / \int_{0.5\mu\text{m}}^{9\mu\text{m}} \eta(\lambda) \cdot A(\lambda) \cdot d\lambda$$

According to the user manual of the global IR source, $\eta(\lambda)$ can be described using the ideal back body radiation at $T = 1500 \text{ K}$ (See Supplementary Figure 8(a) in Supplementary Note 6). Then the optical power arriving at the detector per wavelength $P_{\text{arrive}}(\lambda_{\text{peak}})$ is found to be $0.377 \mu\text{W}/\mu\text{m}$.

It would be nice if the authors addressed these points as it would be misleading to the reader, i.e. led to believe the manuscript presents a full NDIR sensor (as claimed in the abstract) better than what is currently available, which clearly is untrue. In reality the NDIR presented here is significantly bulkier and probably way more expensive than most current NDIRs. The authors must be clear about what is the exact novelty here, i.e. multigas sensing with the plasmonic detector only (not even the detector itself).

In response to Reviewer 3:

The paper presents a novel integrated array of thermal photodetectors, independently addressed, each using a different geometry (size and lattice constant) of nanostructured antennae. The photodetectors thereby become narrowband absorbers suitable for use in gas detection via nondispersive infrared sensing. The level of integration achieved using this technique is a step forward, novel and interesting. I have no doubt that the device construction and attention to detail in its modelling is fine work, and it is an achievement for all the antennae on one device to have their absorption spectra line up so nicely with all the identified gases. This means the optical modelling and level of fidelity achieved during fabrication must have been of high quality.

However, I do have some scientific reservations about the paper, as currently presented. These mainly relate to the instrumentation aspects of the work, i.e. to its design and use as a gas sensor, the experiments performed, and the analysis of the results. On this basis I would conclude that the paper needs a serious rewrite before it is ready for publication in a journal such as this. Given that the authors may currently be unable to access their lab, some of the claims in the paper may have to remain unsubstantiated (for example the measurement of the 3dB point in the frequency response) and the discussion / conclusions should avoid overstepping their experimental basis. My other comments are given below:

Scientific

1. A comment on the first reviewer's point about lithium tantalite. As a reminder, this was the discussion between reviewer and authors:

[Reviewer] The authors fabricated a pyroelectric detector using lithium tantalite as the active layer. The Reviewer is familiar with the pyroelectric perovskite lithium tantalate but was unable to find references to lithium tantalite. The detector structure itself is very timely; pyroelectric-plasmonic detectors are becoming a hot research area, and the structure itself is appropriate for the journal.

[Authors] Lithium tantalite(LT) is one of a large number of octahedral ferroelectrics. Its structure is similar to lithium niobate (LN).

I note that the supplementary material includes reference to the material LiTaO₃, which is conventionally referred to as lithium tantalate. Therefore I assume that there is either a typo in the main text (it should be lithium tantalate) or in the equation in the supplementary data. The supporting references provided by the authors show that we might be talking about lithium tantalate all along.

We thank the reviewer very much for catching the typo. It should be lithium tantalate and we have

corrected this typo in the maintext and the supplementary data.

2. Page 3: lithium tantalate (LT) is described as a good IR absorber, but the effect of occluding this with a gold electrode has not been discussed. Later it is stated that the nanodisk antennae on top convert the IR radiation to heat and the LT material is described as being a heat absorber (presumably via conduction not radiation) therefore the IR absorbing properties of LT are irrelevant. This section therefore needs to be rewritten because at present it is inconsistent. The reason for the choice of LT as sensor material needs to be made clear. Surely the main criterion should be that the material has a high conversion of heat to electrical current.

LT is a typical pyroelectric material that offers a very broadband infrared response, enough to cover the characteristic absorption bands of typical gases. Besides, it has a high conversion efficiency from heat to pyroelectric current. In commercial LT based pyroelectric detectors, the pre-deposited top and bottom gold electrodes form a capacitor structure and convert the generated pyroelectric current to output voltage. To improve the IR absorption, commercial detectors also implement broadband absorber such as metal black on top of the top gold electrode. In this configuration, although the LT layer is occluded by the top gold electrode, the IR absorption of the detector is still high.

In our structure, we replace the broadband metal black absorber with the narrowband MIM absorber to selectively enhance the optical absorption in the target spectral band. The top gold electrode of the LT substrate is also used as the gold backplate of the MIM absorbers for simplicity. The optical energy that is resonantly absorbed by the MIM absorber is dissipated as heat via the free electron absorption in the gold nanodisk antennae. The generated heat is conducted to the LT layer via the gold backplate / top electrode to induce the pyroelectric current.

3. The absorption spectra show nice targeting of the different gases under consideration. One gas I would especially expect a strong cross-response to is water vapor (humidity) in the 5-7 μ m region, which will make the measurement of formaldehyde and NO₂ especially challenging. As human experimenters are a major source of variable concentrations of water vapor in the lab, this will also make measurement of sensor performance challenging as well.

We agree with the reviewer that the absorption of the water vapor in the 5 - 7 μ m region is an important source of cross-response. Therefore we took several measures to minimize the interference from the water vapor. First, the NDIR system is installed with a pump that can take out the air and water vapor inside the gas pipelines. Second, we also put desiccants at the outlets of the pipelines to remove water vapor from the ambient. Third, the NDIR system is located on an optical table that is covered by a hood. The hood can help isolate the water vapor from the human operator. Finally, the measurement lab is well ventilated to minimize the remaining water vapor in the room.

4. An additional comment, worthy of note in the early sections on design and in the discussion of results, is this. I am not sure why the authors omitted a reference photodetector, as typically used in gas sensing via NDIR. Typically, a narrowband detector centered on 3.90 μ m or 3.95 μ m is used for this purpose, as it shows minimal cross-response to many gases. It accounts for broad spectrum variability in the light source intensity. Without it, device performance will be affected.

We agree with the reviewer that in a commercial NDIR sensor such as CO₂ sensor, a dual-pixel detector is fitted with a pair of optical filters so that one filter is centered on 4260 nm as the detection channel and the other on 3900 nm as the reference channel. The concentration of CO₂ can be measured from the ratios of the output voltages of the two pixels. Measuring errors caused by dust or diminishing

radiation intensity are removed by the use of the reference channel.

While in our system, due to the high insertion loss caused by the gas cell, the power of the light that passes through the gas cell and arrives at the narrowband detector is on the order of μW and the corresponding output voltage is only a few mV. Thus we use a reflective objective to focus all the light onto the antenna array of the narrowband detector to ensure the output voltage is measurable. It is difficult to add a reference channel since this necessitates the split of the light in the gas cell and further reduce the optical power in the detection channel. Therefore we only implement the detection channel in the current system.

5. The difference between the calculated and measured voltage response is over an order of magnitude. The authors propose that this is a result of non-vacuum packaging; presumably then, over 90% of the device heat is lost to convection within a timescale more rapid than the modulation frequency of 4Hz. 90% sounds rather large, especially as convection within a small cell will be hampered. Some additional evidence would be needed to substantiate this claim.

We agree with the reviewer that convection is not the only cause of the difference between the calculated and measured voltage response. Other causes include:

- 1) In the calculation, the LT is assumed to be supported by four Si posts that sit on the substrate (heat sink). The numerical model does not include the electrical connections of the LT element. While in the experiment, the LT element was directly mounted on the printed circuit board containing the impedance matching circuit. Silver pastes are applied between the bottom electrode of the LT element and the pins on the PCB to ensure good electrical connection. Also, the top electrode of the LT element is wire bonded to the PCB for electrical connection. Thus, the heat conduction between the LT element and the PCB board of the packaged detector should be more significant than the simulated case.
- 2) In the calculation, the area size of the heat source A_s in the equation $i_p = pA_s \frac{\Delta T_P}{dt}$ is set to be the area size of the LT element (3.6mm^2). While in the measurement, the spot size of the optical beam arriving at the LT element is about 0.8 mm. Therefore the size of the area heated up by the beam is only $\pi/4 \times 0.8^2 \text{mm}^2 = 0.5024 \text{mm}^2$. This is another source of the difference between the calculated and measured voltage response.

6. The 3dB frequency is measured relative to 4Hz, which is the lowest frequency supported by the chopper rather than the frequency corresponding to peak response. Inspection of supplementary Fig 5 confirms this. This makes the measured 3dB point arbitrary, or irrelevant. The experiment on frequency response therefore can't be used in support of any particular 3dB point. To measure this properly, a chopper would be needed that supported lower frequency operation, going sufficiently low to cover the frequency of peak response. They are commercially available, but I assume the authors don't have one.

We do not have the chopper that supports frequency operation lower than 4Hz. As suggested by the reviewer, we removed the term "3dB point" in the revised manuscript.

7. I don't understand why the calculated Johnson noise isn't white. The calculation isn't explained – which resistor is generating the Johnson noise? Most typically, the limiting Johnson noise will be produced by the large transimpedance in a current-to-voltage amplifier, if that is what is being used. Without a circuit diagram for the first stage amplification it is impossible to comment. A circuit

diagram (actual or equivalent major elements) should be provided. The authors must have some idea of what it is (even though they use a commercial amplifier) since they have been able to calculate the Johnson noise themselves.

The calculated Johnson noise is not white because the LT element is a capacitive structure with loss resistance, instead of a pure resistive structure like bolometer.

Here are the related discussions from Page 54, Section 5.4 Johnson Noise, in Chapter 5: Pyroelectric Arrays, of the book [Kruse, Paul W. Uncooled thermal imaging: arrays, systems, and applications. Vol. 51. SPIE press, 2001.]:

In resistive materials, Johnson noise is “white,” i.e., independent of electrical frequency. In capacitive materials, including pyroelectric materials, Johnson noise is associated with the loss resistance. That noise is not white; it exhibits a frequency dependence due to the product of the loss resistance and the capacitance.

For more details about the calculation of the frequency dependent Johnson noise of a pyroelectric detector, please refer to the following application note from <https://www.infratec-infrared.com/>: https://www.infratec-infrared.com/downloads/en/sensor-division/application-notes/application_detector_basics.pdf

Supplementary Figure 6 | The circuit diagram of the packaged narrowband LT detector

The Supplementary Figure 6 in Supplementary Note 5 shows the circuit diagram of the packaged narrowband LT detector. The LT detector is modelled as a parallel plate capacitor C_d with a loss resistance R_d . The generated pyroelectric current is modelled as a current source. And the output of the LT element is coupled to the input of an amplifier, characterized by an input resistance R_G , which is in parallel with the loss resistance. R_s and C_s are load resistance and capacitance. Relevant parameters are:

$$C_d = \epsilon_0 \epsilon_r A_S / t_p = 8.854 \times 10^{-12} \text{ F/m} * 54 * 3.6 \text{ mm} / 75 \text{ um} = 22.95 \text{ pF}$$

$$R_d = 80 \text{ G}\Omega$$

$$R_G = 1 \text{ T}\Omega$$

$$C_s = 0.1 \text{ uF}$$

$$R_s = 47 \text{ k}\Omega$$

$$R = \frac{1}{R_G} + \frac{1}{R_d} \approx 80 \text{ G}\Omega$$

8. On the use of the modified Beer Lambert Law, equation (1)

I don't understand the reasoning behind accounting for diffraction of the light beam using parameter c , since this should be unaffected by the presence of gas (the gases have a relatively small contribution to the refractive index of the air and therefore changes to beam width with concentration should be negligible). I am uncertain as to where the authors got equation (1) from. It is not in their reference [31] (Barnes JR, et al. A Femtojoule Calorimeter Using Micromechanical Sensors. Rev Sci Instrum 65, 3793-3798 (1994)) nor in the associated erratum for that paper.

We first thank the reviewer for pointing out the incorrect reference [31].

The equation (1) was obtained from: Robert Lee and Walt Kester, Analog Dialogue 50-10, October 2016

The incorrect reference [31] has been replaced by the following references:

[32] Robert Lee and Walt Kester, Analog Dialogue 50-10, October 2016
<https://www.analog.com/media/en/analog-dialogue/volume-50/number-4/articles/volume50-number4.pdf>

[33] Application Note 1, A Background to Gas Sensing by Nondispersive Infrared. SGX Sensortech, 2007.
<http://www.sgxsensortech.com/content/uploads/2014/08/AN1---A-Background-to-Gas-Sensing-by-Non-Dispersive-Infrared-NDIR.pdf>

[34] Application Note AAN 201-06, NDIR: Gas Concentration Calculation Overview. Alphasense Limited, 2014
http://www.alphasense.com/WEB1213/wp-content/uploads/2014/12/AAN_201-06.pdf

[35] Application Note AAN 203-04, NDIR: Determination of Linearisation and Temperature Correction Coefficients. Alphasense Limited, 2009.
http://www.alphasense.com/WEB1213/wp-content/uploads/2013/07/AAN_203-04.pdf

[36] Application Note AAN 204-02, NDIR: Origin of Nonlinearity and SPAN. Alphasense Limited, 2009.
http://www.alphasense.com/WEB1213/wp-content/uploads/2013/07/AAN_204-02.pdf

$$\Delta v/v_0 = span * (e^{-\kappa l x^c} - 1) \quad (1)$$

Equation (1) in the maintext is a modified version of the Beer-Lambert Law, as required by the practical considerations in the NDIR implementation.

The parameter $span$ is introduced because not all the IR radiation that impinges upon the detector is absorbed by the gas, even at high concentrations. $span$ is less than 1 because of the optical filter

bandwidth and the fine structure of the absorption spectra.

The parameter c is added into the power term as a linearization coefficient to account for the variations in the optical path length and light scattering for accurately fitting the equation to the actual absorption data.

In practice, the parameters $span$ and c are fitting parameters that are adjusted to match the fitting curves to the measured data as close as possible.

More details about the modified Beer-Lambert law and the use of the parameters such as $span$ and c can be found in the updated references listed above.

5m would be a very long gas cell to have in the lab, was this a multipass cell? It should be described. I suspect it may be a typo. Fig 5(a) does not show a collection lens to collimate the beam from the globar before passing through the cell. I would struggle to measure the intensity of a globar with a thermal detector of the small size indicated here, at a distance of 5m.

The gas cell used in this work is a White type multipass cell with an effective optical length of 5m.

The Supplementary Figure 10(a) in Supplementary Note 9 illustrates the arrangement of the components in the NDIR system, including the Globar light source with the built-in beam collimating capability, the optical chopper, the multipass gas cell, the reflective objective and the narrowband detector that is mounted on an x-y-z translation stage. The Supplementary Figure 10(b) shows the optical configuration of the White cell.

Supplementary Figure 10 | (a) The arrangement of the components in the NDIR system. (b) Basic optical configuration of the White cell.

More details about the White multipass cell can be found in the following references:

[1] John U. White, "Long Optical Paths of Large Aperture," *J. Opt. Soc. Am.* 32, 285-288 (1942), doi:10.1364/JOSA.32.000285

[2] Claude Robert, "Simple, stable, and compact multiple-reflection optical cell for very long optical paths," *Appl. Opt.* 46, 5408-5418 (2007), doi:10.1364/AO.46.005408.

The broadband IR source (SLS203L/M from Thorlab) used in this work is a silicon carbide Globar light source. The Globar is placed inside the housing, it is placed at one focus of an ellipsoid reflector. The output from the ellipsoid reflector is then collimated again with a CaF₂ collimating lens. The Supplementary Figure 11 gives an outline of the optical configuration of the SLS203L/M Globar light source.

[redacted]

Supplementary Figure 11 | Basic Optical Configuration of SLS203L/M

The Supplementary Figure 12 shows the typical angular distribution of the output beam from a SLS203L/M light source.

[redacted]

Supplementary Figure 12 | Angular Distribution of SLS203L/M

More details about the output beam properties of SLS203L/M can be found at:

https://www.thorlabs.com/_sd.cfm?fileName=CTN002679-D02.pdf&partNumber=SLS203L/M

The value of κ required in equation (1) is a composite: the integration of κ values for individual absorption lines over the spectral region that is covered by the detector. The integration introduces an additional nonlinearity, over and above that of the Beer Lambert law, since each of the individual gas lines obeys its own Beer-Lambert equation and becomes nonlinear at a different concentration. This needs to be accounted for as it is important for the later comparison of experiment with theory. The concentrations used are relatively high, the pathlength is long and the results themselves indicate that measurements were made in the nonlinear regime of the Beer-Lambert law where this additional nonlinearity becomes significant.

The authors should modify equation (1) accordingly, and the analysis of data using this equation. Either that or the authors must provide a robust scientific justification for the use of this equation, along with the value of c that applies to their instrument.

In the revised manuscript, the value of k in equation (1) is calculated as followed:

$$\tau = \frac{I}{I_0} = \frac{\int_{\lambda_{min}}^{\lambda_{max}} S(\lambda)T(\lambda)k(\lambda)d\lambda}{\int_{\lambda_{min}}^{\lambda_{max}} S(\lambda)T(\lambda)d\lambda} = e^{-k_{eff}l_0x_0} \quad (S8)$$

and

$$k_{eff} = \frac{-\ln(\tau)}{l_0 x_0} \quad (S9)$$

Here k_{eff} represents the effective absorption coefficient of the target gas; l_0 is the optical length; x_0 is the gas concentration. The integration is from $\lambda_{min} = 0.5 \mu\text{m}$ to $\lambda_{max} = 9 \mu\text{m}$.

$s(\lambda) = \frac{2\pi hc^2}{\lambda^5} \frac{1}{e^{hc/\lambda KT}-1}$ describes the spectral power of radiation from the global IR source. It is the radiated power per unit area of emitting surface per unit wavelength at temperature T . h is the planck constant; c is the speed of light and K is the boltzman constant.

$T(\lambda) = T_0 + \left(\frac{2A}{\pi}\right) \frac{w}{4(\lambda-\lambda_0)^2+w^2}$ describes the spectral response of the narrowband detector. It is the lorentz fitting curve of the spectral absorption of the MIM absorber, where T_0 is an offset value, λ_0 is the center wavelength of spectral absorption, w is the fullwidth at half the maximum, A is the area.

$k(\lambda)$ represents the individual absorption lines of the target gas, extracted one by one from the HITRAN database www.spectraplot.com.

The calculated values of k_{eff} for the eight target gases are shown in the Supplementary Table 4 in Supplementary Note 10:

More details about the calculation of k_{eff} can be found in the following references:

1. S. E. Aleksandrov, G. A. Gavrilov, A. A. Kapralov, B. A. Matveev, G. Y. Sotnikova, and M. A. Remennyi, "Simulation of characteristics of optical gas sensors based on diode optopairs operating in the mid-IR spectral range," Technical Physics **54**, 874-881 (2009).
2. S. Alexandrov, G. Gavrilov, A. Kapralov, S. Karandashev, B. Matveev, G. Sotnikova, and N. Stus, "Portable optoelectronic gas sensors operating in the mid-IR spectral range ($\lambda = 3-5 \mu\text{m}$)," Proceedings of SPIE - The International Society for Optical Engineering **4680**(2002).

9. The mixed gas experiment is welcome, but confusing. The voltages should all be referenced to zero conditions, ie V0 should be measured in nitrogen and not with the fixed gas concentration. I'm not at all sure whether this has been done. Fig 6(a) shows the level of cross-response that one might expect from the original plots of absorbance of the MIMs. Fig 6(b) shows a much more ambiguous response. If one were given the voltage on each channel, would one be able to work out the concentrations of each of the gases? It's not clear from the data in Fig 6(b). That is why I wonder whether the voltages were not referenced to zero gas.

For this section to be convincing, the authors should turn the problem around and see whether their model allows them to accurately work out the concentrations from the detected voltages (and test this on the data).

To turn the problem around and see whether our model allows the calculation of the gas concentrations \mathbf{x}_1 and \mathbf{x}_2 from the detector responses \mathbf{D}_1 and \mathbf{D}_2 , we first write a computer program based the following mathematical model.

$$D_1 = span_{11} * (e^{-k_{11}lx_1^{c11}}) + span_{12} * (e^{-k_{12}lx_2^{c12}}) \quad (S10)$$

$$D_2 = span_{21} * (e^{-k_{21}lx_1^{c21}}) + span_{22} * (e^{-k_{22}lx_2^{c22}}) \quad (S11)$$

The values of the coefficient $span_{ij}$, k_{ij} , c_{ij} are listed in the following table:

Configuration of the experiment	Value of i and j	$span_{ij}$	k_{ij}	c_{ij}
Detector I Fixed SO ₂ , changed CO	i = 1, j = 1	0.31	2.69587×10^{-7}	0.99
Detector I Fixed CO, changed SO ₂	i = 1, j = 2	0.99	3.89657×10^{-8}	0.84
Detector II Fixed SO ₂ , changed CO	i = 2, j = 1	0.99	3.56494×10^{-8}	0.86
Detector II Fixed CO, changed SO ₂	i = 2, j = 2	0.23	1.30559×10^{-6}	0.86

Supplementary Table 5 | The parameters used to draw the red dashed lines in Fig. 6(b)

We then define $\Delta D_1 = |D_{1\text{-calculated}} - D_{1\text{-measured}}|$ and $\Delta D_2 = |D_{2\text{-calculated}} - D_{2\text{-measured}}|$, where $D_{1\text{-calculated}}$ and $D_{2\text{-calculated}}$ are the calculated detector responses, and $D_{1\text{-measured}}$ and $D_{2\text{-measured}}$ are the measured detector responses. Therefore, ΔD_1 and ΔD_2 represent the absolute difference between the the calculated response and measured response of the two detectors. We further define $\Delta D = \sqrt{\Delta D_1^2 + \Delta D_2^2}$ as the standard deviation of ΔD_1 and ΔD_2 .

To work out the best values of the gas concentrations x_1 and x_2 for the given $D_{1\text{-measured}}$ and $D_{2\text{-measured}}$, the program initiates a two-level iteration that varies both x_1 and x_2 from 1 ppm to 12500ppm, with a step size of 2 ppm. In each iteration, the program finds the calculated detector responses $D_{1\text{-calculated}}$ and $D_{2\text{-calculated}}$ from the combinations of x_1 and x_2 using Equation S1 and S2, and then calculate ΔD_1 , ΔD_2 and ΔD respectively. Finally, the program selects the combination of x_1 and x_2 that corresponds to the minimized ΔD (ΔD_{\min}) as the best values for the given $D_{1\text{-measured}}$ and $D_{2\text{-measured}}$.

We select five cases from the mixed gas experiments, as shown by the measured data points A, B, C, D, and E in Supplementary Fig. 13 in Supplementary Note 11, to implement the program. The experimental input gas concentrations, the measured detector responses, the minimized ΔD (ΔD_{\min}), and the calculated gas concentrations are listed in Supplementary Table 6 in Supplementary Note 11.

Ideally, if all the measured detector responses (purple squares) strictly follow the red dash lines in Supplementary Figure 13, the calculated concentrations shall be the same as the experimentally used concentrations. However, it is seen that there are discrepancies between the calculated concentrations and the experimental concentrations. For example, case A in Supplementary Figure 14, the experimental concentrations are $x_1=10000$ ppm and $x_2=7500$ ppm, while the calculated concentrations are $x_1 = 10645$ ppm and $x_2 = 6691$ pm. The discrepancies arise from the fact that the red dash lines in Supplementary Figure 13 generated by equation S10 and S11 do not perfectly match the measured data points (detector responses). The physical causes of this mismatch include the instability of the output power from the IR source, the temperature drift of the detector responses and other changes in the measurement conditions during the NDIR experiments. We expect that by improving the measurement conditions of the NDIR experiment to ensure that the measured detector responses exactly match the red dash lines, the program can accurately work out the gas concentrations from the measured detector responses.

Supplementary Figure 13 | The detector responses of four mixed gas experiments: fixed SO_2 concentration and varying CO concentration measured by detector I; fixed CO concentration and varying SO_2 concentration measured by detector I; fixed SO_2 concentration and varying CO concentration measured by detector II; fixed CO concentration and varying SO_2 concentration measured by detector II, respectively. (in the maintext Fig. 6(b))

Experimental concentrations			Measured responses and minimized ΔD					Calculated concentrations	
Gas mixture	CO x_1 ppm	SO_2 x_2 ppm	$D_{1\text{-measured}}$	ΔD_1	$D_{2\text{-measured}}$	ΔD_2	ΔD_{min}	CO x_1 ppm	SO_2 x_2 ppm
A	10000	7500	-0.25721	0.00378	-0.21556	0.00537	0.00657	10645	6691
B	7500	10000	-0.2379	0.00766	-0.23291	0.00378	0.00854	7943	10337
C	12500	7500	-0.2919	0.01471	-0.20499	0.02568	0.02959	12499	5735
D	7500	12500	-0.25244	0.01352	-0.26633	0.02514	0.02854	8935	12499
E	7500	5000	-0.23335	0.02194	-0.16414	0.01768	0.02818	9653	3453

Supplementary Table 6 | The parameters of gas mixture A,B,C,D,E

Another way to determine the values of x_1 and x_2 for the given $D_{1\text{-measured}}$ and $D_{2\text{-measured}}$, is to set a certain error range of ΔD_1 and ΔD_2 , and search for the combinations of x_1 and x_2 whose corresponding ΔD_1 and ΔD_2 are within the error range. For example, Supplementary Figure 14 plots the color map of ΔD as a function of x_1 and x_2 with the corresponding $\Delta D_1 \leq 0.015$ and $\Delta D_2 \leq 0.015$. Each of these combinations of x_1 and x_2 in the color map can be regarded as a reasonable choice for the computer deduced x_1 and x_2 .

Supplementary Figure 14 | The color map of ΔD as a function of x_1 and x_2 with the corresponding $\Delta D_1 \leq 0.015$ and $\Delta D_2 \leq 0.015$ for **a**, gas mixture A, and **b**, gas mixture B. The red pentagram indicates the experimentally used x_1 and x_2 . The white patch indicates the combinations of x_1 and x_2 with the corresponding $\Delta D \rightarrow 0$

10. Reviewer 1 commented that not enough gas testing had been completed. In my view it is not necessary to perform measurements on all gases at once, to characterize the cross-response pairwise across the range would be adequate, limiting the measurements to those gases that are neighbors on the spectrum or where MIMs are physical neighbors, to show a worst case.

We agree with the reviewer that the measurement of two gases that are neighbors on the spectrum should be adequate. Unfortunately, the experimental work at our institution is still hampered by the COVID-19 pandemic. We therefore use the two gases (CO and SO₂) in the first submission to model the NDIR experiment on two target gases measured by two narrowband detectors.

11. The question of whether there is thermal cross-talk between the MIMs remains unanswered, but is important for device integration, having been previously shown by others (using different technology) to be a limiting concern. Is the cross-talk entirely predicted by the optical overlap between nanodisk absorption spectra, or is there additional cross-talk to be accounted for?

In our proposed architecture, each MIM area corresponds to a narrowband detection element. When multiple MIMs are built onto the same LT substrate, each MIM absorbs a certain portion of the incident light and dissipates the absorbed optical energy into heat. Thus the heat generated in one MIM area can affect the temperature of the neighbouring MIMs via thermal conduction and this is the thermal cross-talk.

To minimize the thermal cross-talk, the basic idea is to thermally isolate each narrowband detection element using its own heat sink. For example, the current 75 μ m thick LT substrate with built-in MIMs can be cutted into separate narrowband detection elements. The separate detection elements can then be mounted on a PCB board via the pins that provide electrical connections, as shown in the Supplementary Figure 15 in Supplementary Note 13. In this case the pins provide heat conduction to the PCB boards and thermally isolate each detection elements.

Supplementary Figure 15 | Packaging and thermal isolation of narrowband detectors using a PCB board

We also plan to fabricate MIMs on 700 nm thick LT thin film on silicon substrate (LTOI). To create thermal isolation, one can fabricate deep trenches in the LT substrate between adjacent MIMs to reduce the thermal conduction, as shown in the Supplementary Figure 16 in Supplementary Note 13.

Supplementary Figure 16 | Thermal isolation of thin film LT based narrowband detector array using laser-cutted trenches

12. Using their knowledge of noise and gas response, the authors should estimate a limit of detection for each gas and the expected level of cross-response for each of the neighboring gases on the spectrum. Limits of detection for some commonly measured gases (eg CO₂ and possibly CO, CH₄) should be provided in the abstract to help others prioritize the paper. The case for “accurate detection” as mentioned in the abstract is not made, and anyway should be quantified: what level of accuracy?

The detection limits of the eight target gases calculated from the measured voltage responses in Supplementary Fig. 6 are summarized in Supplementary Table 3. We calculate the gas detection limit C_{limit} using the following equation:

$$C_{limit} = \frac{3\sigma\Delta C}{\Delta V} \quad (S12)$$

where σ is the voltage deviation at 0 ppm (100% nitrogen atmosphere), and the $\Delta V/\Delta C$ is the sensitivity of the detector at low concentrations.

GAS	H ₂ S	CH ₄	CO ₂	CO	NO	CH ₂ O	NO ₂	SO ₂
-----	------------------	-----------------	-----------------	----	----	-------------------	-----------------	-----------------

Detection Limit [ppm]	489	63	2	11	17	27	54	104
-----	----	---	----	----	----	----	-----

Supplementary Table 7 | The detection limit of eight different gases

The level of cross-response for each of the neighboring gases on the spectrum is evaluated using $X_{MIN} = \frac{1}{k_{effl}} \ln\left(\frac{V_0}{V_0 - \Delta V_{MIN}}\right)$, where $\Delta V_{MIN} = 3 * \sigma$. The calculate values of X_{min} for detector I to VIII are summarized in the Supplementary Table 8 in Supplementary Note 12. For each detector, the X_{min} of its own target gas and the neighboring gases are provided. For example, for detector III that is used for CO₂, the X_{min} of CH₄, CO₂, and CO are provided.

Detector	I (for H ₂ S)		II (for CH ₄)			III (for CO ₂)			IV (for CO)		
Gas	H ₂ S	CH ₄	H ₂ S	CH ₄	CO ₂	CH ₄	CO ₂	CO	CO ₂	CO	NO
X_{min} (ppm)	3089	322	30114	89	106	149	4	28	41	121	806
Detector	V (for NO)		VI (for CH ₂ O)		VII (for NO ₂)		VIII (SO ₂)				
GAS	CO	NO	CH ₂ O	NO	CH ₂ O	NO ₂	CH ₂ O	NO ₂	SO ₂	NO ₂	SO ₂
X_{min} (ppm)	101	50	19	81	13	7	89	18	381	54	9

Supplementary Table 8 | The level of cross-response for each of the neighboring gases on the spectrum

13. I generally agree with the discussion of the device's potential in imaging and on-chip spectroscopy. However, its performance for NDIR sensing is not yet evidenced. The authors have not estimated a limit of detection, but I would be surprised if this was comparable to conventional devices. The main limitation seems to be the small absorber area, which is a natural consequence of the high level of integration and desire to contain all this sensing capability within a small device. The authors should therefore provide additional comment on the prospects for improving the limit of detection while maintaining this integration, if they want this to be a potential application.

The detection limits of the eight target gases calculated from the measured voltage responses are summarized in Supplementary Table 7.

The Supplementary Figure 4 in Supplementary Note 3 presents the optical microscope image of 16 MIM absorbers fabricated on top of a 75um thick LT substrate. Windows for wire-bonding and electrical connection are also created for each MIM absorber area. The area size of each absorber in this work is 1x1 mm. But it can be easily expanded to 2x2 mm, 5x5 mm or even larger. The MIM absorbers can be patterned on 6 inch LT substrate using UV lithography (stepper). The narrowband detectors can be cut and packaged either as a group of single pixel detectors or as a multi-pixel detector.

Supplementary Figure 4 | A group of 16 MIM absorbers fabricated on top of a 75um thick LT substrate

Supplementary Figure 17 presents an example of three gas NDIR sensor based on multiple pairs of bandpass filter and detector¹. Each pair of bandpass filter and detector is TO packaged and mounted on a circuit board. Our narrowband detectors can replace the multiple pairs of bandpass filter and detector in this module and form a multi-gas sensor. Thus the size of the absorber area should not be the main limitation.

Fig. 3. Distribution of detectors and IR source.

Fig. 4. Optical paths in gas chamber.

Supplementary Figure 17 | An example of three gas NDIR sensor based on multiple pairs of bandpass filter and detector.

As the prospects for improving the limit of detection while maintaining this integration, the two key factors are: 1) The thickness of the LT substrate that determines the photothermal responsivity of the narrowband detectors. 2) The quality factor of the MIM absorber that determines the overlap between the spectral response of the detector and the gas absorption lines. There is an on-going effort in our group to fabricate narrowband detectors using 700 nm thick LT thin film, though the progress is delayed by the COVID-19 pandemic. Hopefully the reduction of the LT thickness by two orders of magnitude will effectively increase the detector responsivity and thus enable a miniaturized multi-gas NDIR sensor.

Presentation

14. Page 2. The performance of NDIR sensors is not “unparalleled”. Sensors based on tunable diode laser spectroscopy offer better sensitivity and selectivity. The reason why NDIR sensors are so important is (a) their low cost compared to such alternatives, and (b) the fact that other, potentially lower cost sensors are ineffective for certain gases such as CO2.

The reviewer is talking about the following paragraph.

“As such, mid-IR spectroscopic gas sensors can be employed to uniquely identify and quantify the presence of substances with unparalleled high sensitivity and selectivity. Non-dispersive infrared (NDIR) spectroscopy is one of mid-IR spectroscopic gas sensors that analyzes gases based on their characteristic absorption wavelengths in the mid-IR caused by their molecular vibrations, which can find profound applications in traced gas sensing, breadth analysis, environmental monitoring, to name a few.”

To clarify, the word “unparalleled” in the above paragraph is used to describe “mid-IR spectroscopic gas sensors” as a whole, rather than NDIR sensors only. Spectroscopic gas sensors based on mid-IR tunable quantum cascade lasers can be categorized as “mid-IR spectroscopic gas sensors” and they also offer very good (“unparalleled”) sensitivity and selectivity.

But to avoid any further misunderstanding, we removed the word “unparalleled ” from the manuscript.

15. I could not open the supplementary dataset. It is conventional for open data to use commonly accessible and stable file formats such as csv.

The supplementary dataset has been provided as csv files.

16. Page 2 “Moreover, this scheme does not allow the real-time monitoring of multiple gases at the same time.” I disagree; it would be perfectly possible to have such multiple filter / detector sensors operating simultaneously if the detector elements are independently addressed.

We removed the word “Moreover, this scheme does not allow the real-time monitoring of multiple gases at the same time.” from the manuscript.

1. Tan Q, *et al.* Three-gas detection system with IR optical sensor based on NDIR technology. *Optics and Lasers in Engineering* **74**, 103-108 (2015).

REVIEWERS' COMMENTS:

Reviewer #3 (Remarks to the Author):

The authors have done a good job of responding appropriately to the comments made by me and other reviewers. I have attached a file looking at each comment. I would still like to see a brief comment within the paper concerning the choice of detection bands - specifically those gases that absorb within the 5-7 μ m region and the use (or not) of a reference channel. Other than that I am happy with the changes and I think they provide a useful addition for readers.

In response to Reviewer #3's reply comments numbered 3-5 from the previous round:

3. The absorption spectra show nice targeting of the different gases under consideration. One gas I would especially expect a strong cross-response to is water vapor (humidity) in the 5-7 μ m region, which will make the measurement of formaldehyde and NO₂ especially challenging. As human experimenters are a major source of variable concentrations of water vapor in the lab, this will also make measurement of sensor performance challenging as well.

We agree with the reviewer that the absorption of the water vapor in the 5 - 7 μ m region is an important source of cross-response. Therefore we took several measures to minimize the interference from the water vapor. First, the NDIR system is installed with a pump that can take out the air and water vapor inside the gas pipelines. Second, we also put desiccants at the outlets of the pipelines to remove water vapor from the ambient. Third, the NDIR system is located on an optical table that is covered by a hood. The hood can help isolate the water vapor from the human operator. Finally, the measurement lab is well ventilated to minimize the remaining water vapor in the room.

These seem reasonable efforts for the test procedure, though water vapour will have an overwhelming effect on absorption in the 5-7 μ m region and will need to be removed to a very high degree. However my comments related more to the future utility of the sensor being used in standard atmospheric conditions. I feel this point ought to be commented on within the paper to avoid misleading other researchers who might not understand the subject as well as this research team.

The following paragraph has been added in to the Discussion section (page 10, in purple color):

Since water vapor (humidity) is a strong source of cross-response in the 5 - 7 μ m, we took several measures to minimize the interference from the water vapor. First, the NDIR system is installed with a pump that can take out the air and water vapor inside the gas pipelines. Second, we also put desiccants at the outlets of the pipelines to remove water vapor from the ambient. Third, the NDIR system is located on an optical table that is covered by a hood. The hood can help isolate the water vapor from the human operator. Finally, the measurement lab is well ventilated to minimize the remaining water vapor in the room. Thus in the future, to make sure that the ~cm long NDIR sensors can accurately measure gases in the 5 - 7 μ m region in standard atmospheric conditions, water vapor need to be removed to a very high degree.

4. An additional comment, worthy of note in the early sections on design and in the discussion of results, is this. I am not sure why the authors omitted a reference photodetector, as typically used in gas sensing via NDIR. Typically, a narrowband detector centered on 3.90 μ m or 3.95 μ m is used for this purpose, as it shows minimal cross-response to many gases. It accounts for broad spectrum variability in the light source intensity. Without it, device performance will be affected.

We agree with the reviewer that in a commercial NDIR sensor such as CO₂ sensor, a dual-pixel detector is fitted with a pair of optical filters so that one filter is centered on 4260 nm as the detection channel and the other on 3900 nm as the reference channel. The concentration of CO₂ can be measured from the ratios of the output voltages of the two pixels. Measuring errors caused by dust or diminishing radiation intensity are removed by the use of the reference channel.

While in our system, due to the high insertion loss caused by the gas cell, the power of the light that passes through the gas cell and arrives at the narrowband detector is on the order of μW and the corresponding output voltage is only a few mV. Thus we use a reflective objective to focus all the light onto the antenna array of the narrowband detector to ensure the output voltage is measurable. It is difficult to add a reference channel since this necessitates the split of the light in the gas cell and further reduce the optical power in the detection channel. Therefore we only implement the detection channel in the current system.

Thank you for this explanation. However, I still think a comment ought to be provided within the paper to point this out. I completely agree that it is difficult to split the light, but that is a problem that affects the field generally. All NDIR detectors would have more light if they didn't have to include a reference. I think in future designs, the team should consider replacing one of the elements covering the 5-7 μm range and making one element a reference detector. Presumably, for simultaneous detection they don't focus all the light onto one element at a time, but across the whole array, so there is little to be lost by having a reference within the array (and an advantage for multi-gas detection to share the reference). The global collimation optics (later: thank you for that) will help randomize the global output such that each element should receive a representative portion of the spectrum.

The following paragraph has been added in to the Discussion section (page 11, in purple color):

Due to the high insertion loss caused by the gas cell, the current NDIR system is not implemented with a reference channel. In future designs of $\sim\text{cm}$ long NDIR sensors, one of the detection elements in the 5-7 μm range can be made as a reference detector. Measuring errors caused by dust or diminishing radiation intensity are removed by the use of the reference channel.

5. The difference between the calculated and measured voltage response is over an order of magnitude. The authors propose that this is a result of non-vacuum packaging; presumably then, over 90% of the device heat is lost to convection within a timescale more rapid than the modulation frequency of 4Hz. 90% sounds rather large, especially as convection within a small cell will be hampered. Some additional evidence would be needed to substantiate this claim.

We agree with the reviewer that convection is not the only cause of the difference between the calculated and measured voltage response. Other causes include:

- 1) In the calculation, the LT is assumed to be supported by four Si posts that sit on the substrate (heat sink). The numerical model does not include the electrical connections of the LT element. While in the experiment, the LT element was directly mounted on the printed circuit board containing the impedance matching circuit. Silver pastes are applied between the bottom electrode of the LT element and the pins on the PCB to ensure good electrical connection. Also, the top electrode of the LT element is wire bonded to the PCB for electrical connection. Thus, the heat conduction between the LT element and the PCB board of the packaged detector should be more significant than the simulated case.
- 2) In the calculation, the area size of the heat source A_s in the equation $i_p = pA_s \frac{\Delta T_p}{dt}$ is set to be the area size of the LT element (3.6 mm^2). While in the measurement, the spot size of the optical beam arriving at the LT element is about 0.8 mm. Therefore the size of the area heated up by the beam is only $\pi/4 \times 0.8^2 \text{ mm}^2 = 0.5024 \text{ mm}^2$. This is another source of the difference between the calculated and measured voltage response.

This is a really helpful answer, thank you. First, I think that the calculation could have predicted the

effect of (2) by using the appropriate area. This was predictable and would have provided better agreement between the two. Second, the effects of (1) should be briefly commented within the paper.

The following paragraph has been added in to the Photoresponse of the narrowband detectors section (page 6, in purple color):

Factors that cause the measured responsivity to be lower than the calculated responsivity include: 1) The heat conduction between the LT element and the printed circuit board containing the impedance matching circuit is more significant than the calculated case. 2) The actual spot size of the optical beam arriving at the LT element can be smaller than the area size of the LT element.